# ON LAYER NORMALIZATION IN THE TRANSFORMER ARCHITECTURE

## ABSTRACT

The Transformer architecture is popularly used in natural language processing tasks. To train a Transformer model, a carefully designed learning rate warm-up stage is usually needed: the learning rate has to be set to an extremely small value at the beginning of the optimization and then gradually increases in some given number of iterations. Such a stage is shown to be crucial to the final performance and brings more hyper-parameter tunings. In this paper, we study why the learning rate warm-up stage is important in training the Transformer and theoretically show that the location of layer normalization matters. It can be proved that at the beginning of the optimization, for the original Transformer, which places the layer normalization between the residual blocks, the expected gradients of the parameters near the output layer are large. Then using a large learning rate on those gradients makes the training unstable. The warm-up stage is practically helpful to avoid this problem. Such an analysis motivates us to investigate a slightly modified Transformer architecture which locates the layer normalization inside the residual blocks. We show that the gradients in this Transformer architecture are well-behaved at initialization. Given these findings, we are the first to show that this Transformer variant is easier and faster to train. The learning rate warm-up stage can be safely removed, and the training time can be largely reduced on a wide range of applications.

## 1 INTRODUCTION

The Transformer is one of the most commonly used neural network architectures in natural language processing, and layer normalization is one of the key components in the Transformer. The originally designed Transformer places the layer normalization between the residual blocks, which is usually referred to as the Transformer with Post-Layer Normalization (Post-LN). This architecture has achieved state-of-the-art performance in many tasks including language modeling (Dai et al., 2019; Al-Rfou et al., 2018) and machine translation (Vaswani et al., 2017; Dehghani et al., 2018; Edunov et al., 2018). Unsupervised pre-trained models based on the Post-LN Transformer architecture also show impressive performance in many downstream tasks (Radford et al., 2019; Devlin et al., 2018; Yang et al., 2019).

Although it achieves great success, people usually need to deal with the optimization of the Post-LN Transformer more carefully than convolutional networks (He et al., 2016; Tan & Le, 2019) or other sequence-to-sequence models (Sutskever et al., 2014; Gehring et al., 2017). In particular, to train the model from scratch, any gradient-based optimization approach requires a learning rate warm-up stage: The optimization starts from using an extremely small learning rate, e.g., $1e^{-7}$, and then gradually increases it to a pre-defined maximum value in a pre-defined number of iterations. After that, the learning rate decays similar to the optimization of other architectures. Both previous works (Vaswani et al., 2017; Popel & Bojar, 2018), as well as our empirical study, show that such a warm-up stage is essential in training the models. Furthermore, the final model performance is quite sensitive to the value of the maximum learning rate and the number of warm-up iterations. Tuning such sensitive hyper-parameters is costly in training large-scale models, e.g., BERT (Devlin et al., 2018) or XLNet (Yang et al., 2019).

In this paper, we study why the learning rate warm-up stage is essential in the optimization of the Post-LN Transformer and find it is closely related to the position of the layer normalization. As

the warm-up stage happens in the first several iterations, we investigate the optimization behavior at initialization of the Post-LN Transformer. According to our theoretical analysis, when putting the layer normalization between the residual blocks, the expected gradients of the parameters near the output layer are large. Therefore, without the warm-up stage, directly using a large learning rate to those parameters may not lead to an improved model and can even make the optimization process unstable. Using a warm-up stage and training the model from small learning rates practically avoid this problem.

As the location of the layer normalization plays a crucial role in controlling the gradient scales, we investigate whether there are some other ways of positioning the layer normalization that lead to better-normalized gradients. In particular, we study another variant, the Transformer with Pre-Layer Normalization (Pre-LN) (Klein et al., 2018). The Pre-LN Transformer puts the layer normalization inside the residual connection and equips with an additional *final-layer normalization* before prediction (Please see Figure 1 for the differences between the two variants of the Transformer architectures). In this paper, we show that the gradients are well-behaved without any exploding or vanishing at initialization for the Pre-LN Transformer both theoretically and empirically.

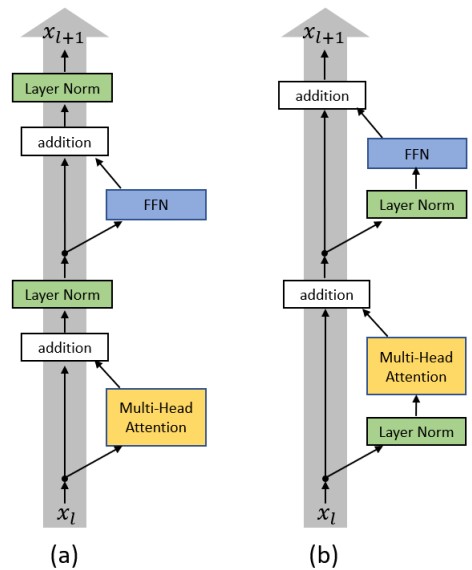

Figure 1: (a) Post-LN Transformer layer; (b) Pre-LN Transformer layer.

Given the gradients are well-behaved in the Pre-LN Transformer, it is natural to consider removing the learning rate warm-up stage during training. We conduct extensive experiments, including IWSLT14 German-English translation, WMT14 English-German translation, and BERT pre-training tasks. We show that, in all tasks, the learning rate warm-up stage can be safely removed and thus, the number of hyper-parameter is reduced. Furthermore, we observe that the loss decays faster for the Pre-LN Transformer model. It can achieve comparable final performances but use much less training time. This is particularly important for training large-scale models on large-scale datasets.

Our contributions are summarized as follows: First, we investigate two Transformer variants, the Post-LN Transformer and the Pre-LN Transformer. By studying the gradients at initialization, we show why the learning rate warm-up stage is essential in training the Post-LN Transformer. Second, we are the first to show that the learning-rate warm-up stage can be removed for the Pre-LN Transformer. By using proper learning rate schedulers, the training time can be largely reduced.

## 2  RELATED WORK

Gradient descent-based methods (Kingma & Ba, 2014; Zeiler, 2012; Duchi et al., 2011; Tieleman & Hinton, 2012) are popularly used in optimizing deep neural networks. For convolutional neural networks and recurrent neural networks, a relatively large learning rate is usually set in the beginning, and then decays along with the optimization process (He et al., 2016; 2017; Sutskever et al., 2014; Gehring et al., 2017; He et al., 2019). The learning rate warm-up stage has been shown to be essential in dealing with some specific problems, e.g., the large-batch training. Goyal et al. (2017); He et al. (2019) and You et al. (2018) showed that when training neural networks with extremely large batch sizes (e.g., 8k in ImageNet), optimizing the model with a large learning rate in the beginning usually leads to poor performance. Training with a learning rate warm-up stage can eliminate the performance gap.

However, when optimizing the Post-LN Transformer models, as far as we know, in almost all previous works (Vaswani et al., 2017; Devlin et al., 2018; Dai et al., 2019; Radford et al., 2019; Lu

et al., 2019), the learning rate warm-up stage is essential and critical for training. Popel & Bojar (2018) investigated the influence of different warm-up strategies for the optimization of the Post-LN Transformer model and found that without or with relatively less warm-up iterations (e.g., 12k in Cz-En translation), the optimization diverges.

In a concurrent and independent work (Liu et al., 2019a), the authors claimed that the benefit of the warm-up stage comes from reducing the variance for the adaptive learning rate in the Adam optimizer (Kingma & Ba, 2014). They proposed to rectify the variance of adaptive learning rate by a new variant of Adam called RAdam. However, we identify the problem from the parameter initialization. We show that for the Post-LN Transformer, the scales of gradients of some parameters at initialization are large. First-order optimizers take the gradients as input. Using such gradients on these optimizers (not limit to Adam) with a large learning rate may make the optimization unstable and hurt the final performance.

## 3 OPTIMIZATION FOR THE TRANSFORMER

### 3.1 THE TRANSFORMER ARCHITECTURE WITH POST-LAYER NORMALIZATION

The Transformer architecture usually consists of stacked Transformer layers (Vaswani et al., 2017; Devlin et al., 2018), each of which takes a sequence of vectors as input and outputs a new sequence of vectors with the same shape. A Transformer layer has two sub-layers: the (multi-head) self-attention sub-layer and the position-wise feed-forward network sub-layer. Residual connection (He et al., 2016) and layer normalization (Lei Ba et al., 2016) are applied for both sub-layers individually. We first introduce each component of the Transformer layer and then present the entire architecture.

**Self-attention sub-layer** An attention function can be formulated as querying an entry with key-value pairs (Vaswani et al., 2017). The self-attention sub-layer uses scaled dot-product attention, which is defined as: $\text{Attention}(Q, K, V) = \text{softmax}(\frac{QK^T}{\sqrt{d}})V$, where $d$ is the dimensionality of the hidden representations, and $Q$ (Query), $K$ (Key), $V$ (Value) are specified as the hidden representations of the previous layer. The multi-head variant of the self-attention sub-layer is popularly used which allows the model to jointly attend to information from different representation sub-spaces, and is defined as

$$\text{Multi-head}(Q, K, V) = \text{Concat}(\text{head}_1, \cdots, \text{head}_H)W^O, \tag{1}$$

$$\text{head}_k = \text{Attention}(QW_k^Q, KW_k^K, VW_k^V), \tag{2}$$

where $W_k^Q \in \mathbb{R}^{d \times d_K}, W_k^K \in \mathbb{R}^{d \times d_K}, W_k^V \in \mathbb{R}^{d \times d_V}$, and $W^O \in \mathbb{R}^{Hd_V \times d}$ are project parameter matrices, $H$ is the number of heads. $d_K$ and $d_V$ are the dimensionalities of Key and Value. Without any confusion, given a sequence of vectors $(x_1, ..., x_n)$, we use $\text{MultiHeadAtt}(x_i, [x_1, x_2, \cdots, x_n])$ as the multi-head self-attention mechanism on position $i$ which considers the attention from $x_i$ to the entire sequence.

**Position-wise FFN sub-layer** In addition to the self-attention sub-layer, each Transformer layer contains a fully connected network, which is applied to each position separately and identically. This sub-layer is a two-layer feed-forward network with a ReLU activation function. Given a sequence of vectors $h_1, ..., h_n$, the computation of a position-wise FFN sub-layer on any $h_i$ is defined as:

$$\text{FFN}(h_i) = \text{ReLU}(h_i W^1 + b^1)W^2 + b^2, \tag{3}$$

where $W^1$, $W^2$, $b^1$ and $b^2$ are parameters.

**Residual connection and layer normalization** Besides the two sub-layers described above, the residual connection and layer normalization are also key components to the Transformer. For any vector $v$, the layer normalization is computed as $\text{LayerNorm}(v) = \gamma \frac{v - \mu}{\sigma} + \beta$, in which $\mu, \sigma$ are the mean and standard deviation of the elements in $v$, i.e., $\mu = \frac{1}{d} \sum_{k=1}^{d} v_k$ and $\sigma^2 = \frac{1}{d} \sum_{k=1}^{d} (v_k - \mu)^2$. Scale $\gamma$ and bias vector $\beta$ are parameters.

Different orders of the sub-layers, residual connection and layer normalization in a Transformer layer lead to variants of Transformer architectures. One of the original and most popularly used

Table 1: Post-LN Transformer v.s. Pre-LN Transformer

| Post-LN Transformer | Pre-LN Transformer |
|---|---|
| $x_{l,i}^{post,1} = \text{MultiHeadAtt}(x_{l,i}^{post}, [x_{l,1}^{post}, \cdots, x_{l,n}^{post}])$ | $x_{l,i}^{pre,1} = \text{LayerNorm}(x_{l,i}^{pre})$ |
| $x_{l,i}^{post,2} = x_{l,i}^{post} + x_{l,i}^{post,1}$ | $x_{l,i}^{pre,2} = \text{MultiHeadAtt}(x_{l,i}^{pre,1}, [x_{l,1}^{pre,1}, \cdots, x_{l,n}^{pre,1}])$ |
| $x_{l,i}^{post,3} = \text{LayerNorm}(x_{l,i}^{post,2})$ | $x_{l,i}^{pre,3} = x_{l,i}^{pre} + x_{l,i}^{pre,2}$ |
| $x_{l,i}^{post,4} = \text{ReLU}(x_{l,i}^{post,3}W^{1,l} + b^{1,l})W^{2,l} + b^{2,l}$ | $x_{l,i}^{pre,4} = \text{LayerNorm}(x_{l,i}^{pre,3})$ |
| $x_{l,i}^{post,5} = x_{l,i}^{post,3} + x_{l,i}^{post,4}$ | $x_{l,i}^{pre,5} = \text{ReLU}(x_{l,i}^{pre,4}W^{1,l} + b^{1,l})W^{2,l} + b^{2,l}$ |
| $x_{l+1,i}^{post} = \text{LayerNorm}(x_{l,i}^{post,5})$ | $x_{l+1,i}^{pre} = x_{l,i}^{pre,5} + x_{l,i}^{pre,3}$ |
| | Final LayerNorm: $x_{Final,i}^{pre} \leftarrow \text{LayerNorm}(x_{L+1,i}^{pre})$ |

architecture for the Transformer and BERT (Vaswani et al., 2017; Devlin et al., 2018) follows "self-attention (FFN) sub-layer $\rightarrow$ residual connection $\rightarrow$ layer normalization", which we call the Transformer with Post-Layer normalization (Post-LN Transformer), as illustrated in Figure 1.

**Post-LN Transformer**   Denote $x_{l,i}$ as the input of the $l$-th Transformer layer at position $i$, where $x_{l,i}$ is a real-valued vector of dimension $d$, $i = 1, 2, ..., n$, $l = 1, 2, ..., L$. $n$ is the length of the sequence and $L$ is the number of layers. For completeness, we define $x_{0,i}$ as the input embedding at position $i$ which is usually a combination of word embedding and positional embedding. The computations inside the $l$-th Post-LN Transformer layer are composed of several steps, and we use super-scripts on $x$ to present the input(output) of different steps as in Table 1 (left), where $W^{1,l}$, $W^{2,l}$, $b^{1,l}$ and $b^{2,l}$ are parameters of the FFN sub-layer in the $l$-th layer.

## 3.2   THE IMPORTANCE OF THE WARM-UP STAGE IN TRAINING THE POST-LN TRANSFORMER

We are interested in the *learning rate warm-up* stage in the optimization of the Post-LN Transformer. Different from the optimization of many other architectures in which the learning rate starts from a relatively large value and then decays (Bahdanau et al., 2017; He et al., 2016; Dauphin et al., 2017), a learning rate warm-up stage for the Post-LN Transformer is **critical**. Specifically, denote the learning rate of the $t$-th iteration as $\text{lr}(t)$. Denote the maximum learning rate during training as $\text{lr}_{max}$. Given a predefined time frame $T_{\text{warmup}}$, the learning rate scheduler for the first $T_{\text{warmup}}$ iterations is defined as (Vaswani et al., 2018)

$$\text{lr}(t) = \frac{t}{T_{\text{warmup}}}\text{lr}_{max}, t \leq T_{\text{warmup}}. \tag{4}$$

After this warm-up stage, the learning rate will be set by classical learning rate schedulers, such as the linear decay, the inverse square-root decay, or forced decay at particular iterations. As we can see from Eqn (4) , at the beginning of the training, the learning rate starts from zero[1] and then linearly increases to $\text{lr}_{max}$ in $T_{\text{warmup}}$ iterations. We conduct experiments to show that this learning rate warm-up stage is essential for training Post-LN Transformer models.

**Setting**   We study the optimization process on the IWSLT14 German-to-English (De-En) machine translation task. We mainly investigate two aspects: whether the learning rate warm-up stage is essential and whether the final model performance is sensitive to the value of $T_{\text{warmup}}$. To study the first aspect, we train the model with the Adam optimizer (Kingma & Ba, 2014) and the vanilla SGD optimizer (Ruder, 2016) respectively. For both optimziers, we check whether the warm-up stage can be removed. We follow (Vaswani et al., 2017) to set hyper-parameter $\beta$ to be $(0.9, 0.98)$ in Adam. We also test different $\text{lr}_{max}$ for both optimizers. For Adam, we set $\text{lr}_{max} = 5e^{-4}$ or $1e^{-3}$, and for SGD, we set $\text{lr}_{max} = 5e^{-3}$ or $1e^{-3}$. When the warm-up stage is used, we set $T_{\text{warmup}} = 4000$ as suggested by the original paper (Vaswani et al., 2017). To study the second aspect, we set $T_{\text{warmup}}$ to be 1/500/4000 ("1" refers to the no warm-up setting) and use $\text{lr}_{max} = 5e^{-4}$ or $1e^{-3}$ with Adam. For all experiments, a same inverse square root learning rate scheduler is used after the warm-up stage. We use both validation loss and BLEU (Papineni et al., 2002) as the evaluation measure of the model performance. All other details can be found in the Appendix.

---

[1]An extremely small learning rate is practically used for the first update, e.g., $1e^{-7}$.

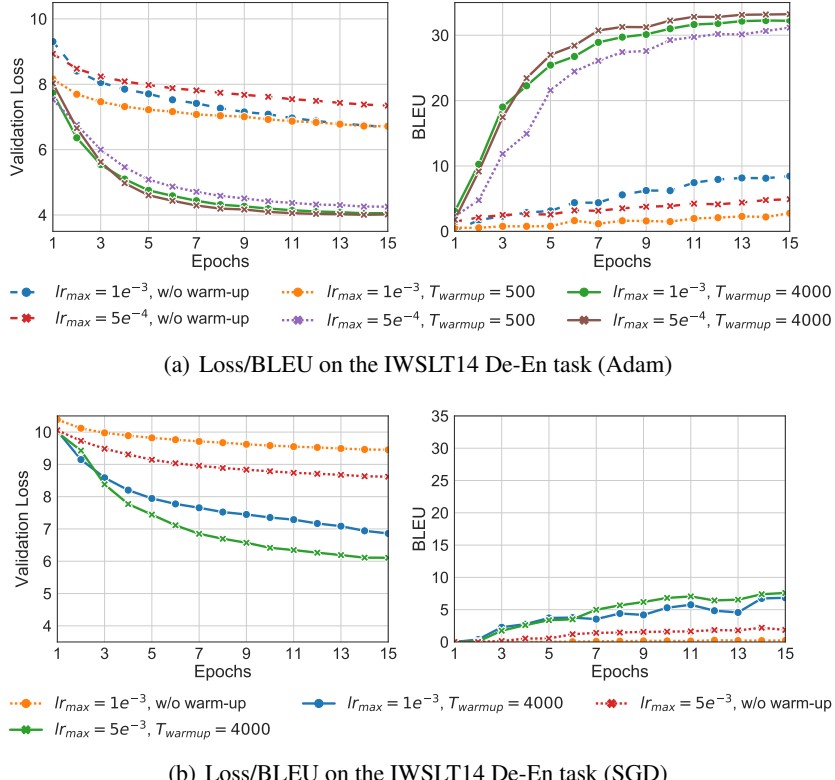

(a) Loss/BLEU on the IWSLT14 De-En task (Adam)

(b) Loss/BLEU on the IWSLT14 De-En task (SGD)

Figure 2: Performances of the models optimized by Adam and SGD on the IWSLT14 De-En task.

**Result**   We record the model checkpoints for every epoch during training and calculate the validation loss and BLEU score. The performance of the models trained with Adam and SGD are plotted in Figure 2(a) and Figure 2(b). The x-axis is the epoch number and the y-axis is the BLEU score/validation loss. "w/o warm-up" indicates "without the warm-up stage" while "w/ warm-up" indicates "with the warm-up stage".

First, we can see that for both optimizers, the learning rate warm-up stage is essential. Without the warm-up stage, the BLEU score of the model trained with Adam optimizer can only achieve 8.45. As a comparison, the model trained using the warm-up stage can achieve around 34 in terms of BLEU score. The same trend can be also observed on the validation loss curves. Although the performance of the model trained with SGD is significantly worse than Adam, we can still see similar phenomena as Adam. The BLEU score is just above zero in 15 epochs without using the warm-up stage.

Second, we can see that the optimization process is sensitive to the value of $T_{\text{warmup}}$, which means $T_{\text{warmup}}$ is an important hyper-parameter in training the Post-LN Transformer. For example, when setting $T_{\text{warmup}} = 500$, the learned models with Adam achieve only 31.16 and 2.77 in term of BLEU score for $lr_{max} = 5e^{-4}$ and $1e^{-3}$ respectively [2] .

**Discussion**   First, we can see that the learning rate warm-up stage significantly helps the optimization of the Post-LN Transformer and also significantly affects the final performance. Such a warm-up stage brings additional efforts on hyper-parameter tuning which is computationally expensive for large-scale NLP tasks. Second, at the beginning of the training, the loss value is usually large. Standard optimization algorithms usually start with a large learning rate for fast convergence. However, when using the warm-up stage, the learning rate has to gradually increase from zero,

---

[2] One may notice that the orange curve is better than the blue curve on the validation loss but is worse on the BLEU score. This is because the BLEU score is defined on the translation results which is generated by step-wise decoding. There may be a big gap between the BLEU score and the validation loss when the model is not well-trained.

which may slow down the optimization process. Liu et al. (2019a) suggests that the warm-up stage plays a role in reducing the undesirably significant variance in Adam in the early stage of model training. Based on this, they design a new variant of Adam optimizer, RAdam. However, according to our results, the warm-up stage also helps the training of SGD. This suggests that the benefit of the warm-up stage may be not for a particular optimizer.

### 3.3 UNDERSTANDING THE TRANSFORMER AT INITIALIZATION

We can see that the Post-LN Transformer cannot be trained with a large learning rate from scratch. This motivates us to investigate what happens at the model initialization. We first introduce the parameter initialization setting for our theoretical analysis and then present our theoretical findings.

**Notations** We denote $\mathcal{L}(\cdot)$ as the loss function of one position, $\tilde{\mathcal{L}}(\cdot)$ as the loss function of the whole sequence, $\| \cdot \|_2$ and $\| \cdot \|_F$ as the $l_2$ norm (spectral norm) and the Frobenius norm, $\text{LN}(x)$ as the standard layer normalization with scale $\gamma = 1$ and bias $\beta = 0$, and $\mathbf{J}_{LN}(x) = \frac{\partial \text{LN}(x)}{\partial x}$ as the Jacobian matrix of $\text{LN}(x)$. Let $\mathcal{O}(\cdot)$ denote standard Big-O notation that suppress multiplicative constants.

**Parameter Initialization** There are multiple parameter matrices in each Transformer layer, and most of the parameter matrices are initialized by the Xavier initialization (Glorot & Bengio, 2010). Given a matrix of size $n_{in} \times n_{out}$, the Xavier initialization sets the value of each element by independently sampling from Gaussian distribution $N(0, \frac{2}{n_{in} + n_{out}})$. The bias vectors are usually initialized as zero vectors. The scale $\gamma$ in the layer normalization is set to one.

For theoretical analysis, we study a simpler setting. First, we focus on single-head attention instead of the multi-head variant and for all layers, we set the shape of $W^{Q,l}$, $W^{K,l}$, $W^{V,l}$, $W^{1,l}$, $W^{2,l}$ to be $d \times d$. Second, we initialize the parameter matrices in the self-attention sub-layer $W^{Q,l}$ and $W^{K,l}$ to be zero matrices. In this setting, the attention is a uniform distribution at initialization and $\text{MultiHeadAtt}(x_{l,i}^1, [x_{l,1}^1, x_{l,2}^1, \cdots, x_{l,n}^1])$ can be simplified as $\frac{1}{n} \sum_{j=1}^{n} x_{l,j} W^{V,l}$. We test the optimization process in this setting and find that the problems described previously remain. Third, we assume the input vectors are also sampled from the same Gaussian distribution. This is reasonable since the inputs to the Transformer are linear combinations of word embeddings and learnable positional embeddings, both of which are initialized by Gaussian distributions.

**Post-LN Transformer v.s. Pre-LN Transformer** We compare the Post-LN Transformer with another variant of the Transformer architecture, the Transformer with Pre-Layer Normalization (Pre-LN). The Pre-LN Transformer was implemented in several systems (Vaswani et al., 2018; Klein et al., 2018; Liu et al., 2019b). Wang et al. (2019) suggested that when stacking more layers, the Pre-LN Transformer is better than its Post-LN counterpart. Different from the Post-LN Transformer that puts the layer normalization between the residual blocks, the Pre-LN Transformer puts the layer normalization inside the residual connection and places it before all other non-linear transformations. Additionally, the Pre-LN Transformer uses a *final layer normalization* right before the prediction. We provide the mathematical formulations and visualizations of the Post-LN Transformer and the Pre-LN Transformer in Table 1 and Figure 1.

For both architectures, each $x_{L,i}$ passes through a softmax layer to produce a distribution over the dictionary $V$. The loss function is defined on the softmax distribution. For example, in sequence prediction, the loss function is defined as $\mathcal{L}(x_{L+1,i}^{post}) = -\log(\text{softmax}_{y_i}(W^{emb} x_{L+1,i}^{post}))$ for the Post-LN Transformer and $\mathcal{L}(x_{Final,i}^{pre}) = -\log(\text{softmax}_{y_i}(W^{emb} x_{Final,i}^{pre}))$ for the Pre-LN Transformer, where $\text{softmax}_{y_i}$ is the probability of ground truth token $y_i$ outputted by the softmax distribution and $W^{emb}$ is the word embedding matrix. The loss of the whole sequence is an average of the loss on each position. Without loss of generality, we assume that all the derivatives are bounded. We introduce the following concentration property of random variables which will be further used in the theorem.

**Definition 1.** *A random variable $Z \geq 0$ is called $(\epsilon, \delta)$-bounded if with probability at least $1 - \delta$, $\frac{Z - \mathbb{E}Z}{\mathbb{E}Z} \leq \epsilon$, where $\epsilon > 0$ and $0 < \delta < 1$.*

Intuitively, if the random variable $Z$ is $(\epsilon, \delta)$-bounded, then with a high probability its realization will not get too far away from its expectation. For example, if $Y$ is a $d$-dimensional standard Gaussian random vector, then $Z = \|Y\|_2^2$ is $(\epsilon, \delta)$-bounded with $\delta = \exp(-d\epsilon^2/8)$, $0 < \epsilon < 1$ (see Appendix G for details). As parameter matrices in self-attention sub-layers and FFN sub-layers are initialized by Gaussian distributions, if the norm of the hidden states in the Transformer satisfies the concentrated condition above, we have the following theorem to characterize the scale of the gradients.

**Theorem 1** (Gradients of the last layer [3] in the Transformer). *Assume that $\|x_{L,i}^{post,5}\|_2^2$ and $\|x_{L+1,i}^{pre}\|_2^2$ are $(\epsilon, \delta)$-bounded for all $i$, where $\epsilon$ and $\delta = \delta(\epsilon)$ are small numbers. Then with probability at least $0.99 - \delta - \frac{\epsilon}{0.9+\epsilon}$, for the Post-LN Transformer with $L$ layers, the gradient of the parameters of the last layer satisfies*

$$\|\frac{\partial \tilde{\mathcal{L}}}{\partial W^{2,L}}\|_F \leq \mathcal{O}(d\sqrt{\ln d})$$

*and for the Pre-LN Transformer with $L$ layers,*

$$\|\frac{\partial \tilde{\mathcal{L}}}{\partial W^{2,L}}\|_F \leq \mathcal{O}\left(d\sqrt{\frac{\ln d}{L}}\right).$$

From Theorem 1, we can see that for the Post-LN Transformer, the scale of the gradients to the last FFN layer is of order $\mathcal{O}(d\sqrt{\ln d})$ which is independent of $L$. For the Pre-LN Transformer, the scale of the gradients is much smaller. We first study the forward propagation of the Post-LN Transformer and the Pre-LN Transformer. Lemma 1 will be served as a basic tool to prove the main theorem and other lemmas.

**Lemma 1.** *If $X \in \mathbb{R}^d$ is a Gaussian vector, $X \sim N(0, \sigma^2\mathbf{I}_d)$, then $\mathbb{E}(\|ReLU(X)\|_2^2) = \frac{1}{2}\sigma^2 d$.*

Based on Lemma 1, we have the following lemma to estimate the scale of the hidden states in different layers for the Post-LN Transformer and the Pre-LN Transformer.

**Lemma 2.** *At initialization, for the Post-LN Transformer, $\mathbb{E}(\|x_{l,i}^{post,5}\|_2^2) = \frac{3}{2}d$ for all $l > 0$ and $i$. For the Pre-LN Transformer, $(1 + \frac{l}{2})d \leq \mathbb{E}(\|x_{l,i}^{pre}\|_2^2) \leq (1 + \frac{3l}{2})d$ for all $l > 0$ and $i$. Expectations are taken over the input and the randomness of initialization.*

Lemma 2 studies the expected norm of the hidden states in both Post-LN/Pre-LN Transformer. It is obviously that in the Post-LN Transformer, the norm of $x_{l,i}^{post}$ is $\sqrt{d}$ and thus we study the norm of $x_{l,i}^{post,5}$ instead. As we can see from Lemma 2, the scale of the hidden states in the Post-LN Transformer keeps to be the same in expectation while the scale of the hidden states in the Pre-LN Transformer grows linearly along with the depth. The next lemma shows that the scale of the hidden states highly relates to the scale of the gradient in the architectures using layer normalization.

**Lemma 3.** *For $x \in \mathbb{R}^d$, we have $\|\mathbf{J}_{LN}(x)\|_2 = \mathcal{O}(\frac{\sqrt{d}}{\|x\|_2})$ in which $\mathbf{J}_{LN}(x) = \frac{\partial LN(x)}{\partial x}$.*

The proof of Lemma 1, Lemma 2, Lemma 3, and Theorem 1 can be found in the Appendix. The main idea is that the layer normalization will normalize the gradients. In the Post-LN Transformer, the scale of the inputs to the layer normalization is independent of $L$, and thus the gradients of parameters in the last layer are independent of $L$. While in the Pre-LN Transformer, the scale of the input to the final layer normalization is linear in $L$, and thus the gradients of all parameters will be normalized by $\sqrt{L}$.

**Extending to other layers/parameters**  We have provided a formal proof on the gradients of the last FFN sub-layer as above. In order to fully understand the optimization, we also make some preliminary analysis for other layers and other parameters. Our main result is that the gradient norm in the Post-LN Transformer is large for the parameters near the output and will be likely to decay as the layer index $l$ decreases. On the contrary, the gradient norm in the Pre- Transformer will be likely to stay the same for any layer $l$. All the preliminary theoretical results are provided in Appendix F.

---

[3]We mainly study the optimization of the Transformer layers and "last layer" here refers to the top FFN layer before the softmax operator. We did not focus on the parameters in the softmax layer as it is tied with the word embedding matrix, which is actually the input to the Transformer.

**Empirical study** We also conduct experiments to study the gradients at initialization for the PostLN/Pre-LN Transformer in real scenarios. The model and training configuration exactly follows Section 3.2. The experiments are repeated ten times using different random seeds. Given an initialized model, we record the hidden states in the Post-LN/Pre-LN Transformer and find that the norm of the hidden states satisfies the concentration property ((0.1,0.125)- bounded). We also record the gradient for each parameter for different mini-batches. For elements in a parameter matrix, we calculate their expected gradients and use the Frobenius norm of those values as the scale of the expected gradient of the matrix. Figure 3(a) and 3(b) shows those statistics for FFN sub-layers. The x-axis indexes different Transformer layers. It can be seen from the figure, the scale of the expected gradients grows along with the layer index for the Post-LN Transformer. On the contrary, the scale almost keeps the same for different layers in the Pre-LN Transformer. These observations are consistent with our theoretical findings. More analysis can be found in Appendix H.

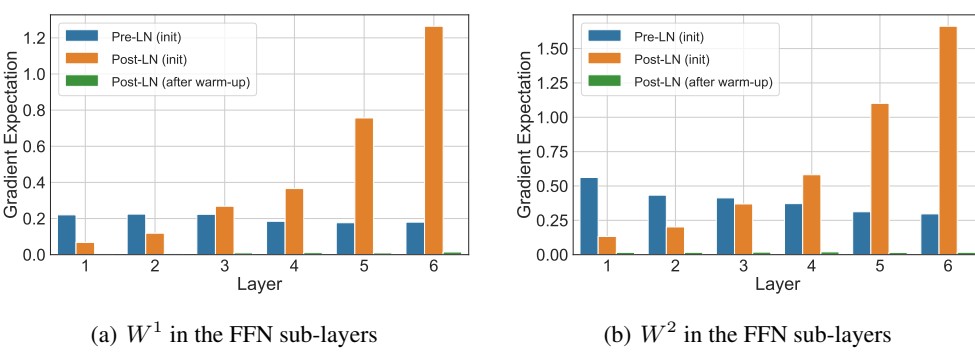

(a) $W^1$ in the FFN sub-layers  (b) $W^2$ in the FFN sub-layers

Figure 3: Norm of expected gradients for Pre-LN/Post-LN Transformer

We further study the gradient statistics for the Post-LN Transformer after the warm-up stage with Adam. It can be seen from the figure that the scale of the gradients are very small, and the model can be trained with large learning rates. We believe the gradient scale is one of the reasons that the Post-LN Transformer needs a careful learning rate scheduling in the beginning. Since the gradients are large for some layers, using a large learning rate without warm-up may make the training unstable (see Appendix I). As the gradients are well-behaved for the Pre-LN Transformer, we will show that the learning rate warm-up stage can be removed for this model architecture in the next section.

## 4 EXPERIMENTS

The Pre-LN Transformer has been implemented in several systems (Liu et al., 2019b; Baevski & Auli, 2018), but most of them still follow Vaswani et al. (2017) to use the learning rate warm-up stage. We conduct experiments for the Pre-LN Transformer to test whether the learning rate warm-up stage can be removed and how to set learning rate schedulers.

### 4.1 EXPERIMENT SETTINGS

**Machine Translation** We conduct our experiments on two widely used tasks: the IWSLT14 German-to-English (De-En) task and the WMT14 English-to-German (En-De) task. For the IWSLT14 De-En task, we use the same model configuration as in Section 3. For the WMT14 En-De task, we use the Transformer base setting. More details can be found in the Appendix.

For training the Pre-LN Transformer, we remove the learning rate warm-up stage. On the IWSLT14 De-En task, we set the initial learning rate to be $5e^{-4}$ and decay the learning rate at the 8-th epoch by 0.1. On the WMT14 En-De task, we run two experiments in which the initial learning rates are set to be $7e^{-4}/1.5e^{-3}$ respectively. Both learning rates are decayed at the 6-th epoch followed by the inverse square root learning rate scheduler.

We train the Post-LN Transformer using the learning rate warm-up stage as the baseline. In both IWSLT14 De-En task and WMT14 En-De task, we set the number of the warm-up stage to be 4000

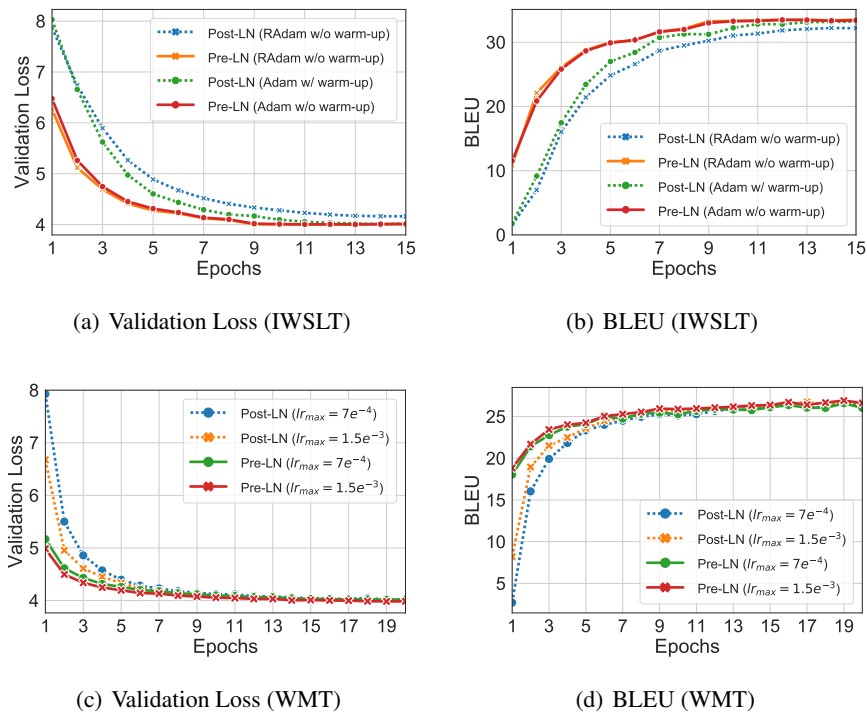

Figure 4: Performances of the models on the IWSLT14 De-En task and WMT14 En-De task

following Vaswani et al. (2017) and then use the inverse square root learning rate scheduler. For all experiments above, we use the Adam optimizer and set the hyper-parameter $\beta$ to be $(0.9, 0.98)$. We set $lr_{max}$ as same as the initial learning rates of the Pre-LN Transformer in each corresponding experiment. Since Liu et al. (2019a) suggests that the learning rate warm-up stage can be removed using RAdam, we try this optimizer on the IWSLT14 De-En task. We use linear learning rate decay suggested by Liu et al. (2019a) and keep all other hyper-parameters to be the same as in other experiments.

**Unsupervised Pre-training (BERT)**   We follow (Devlin et al., 2018) to use English Wikipedia corpus and BookCorpus for pre-training. As the dataset BookCorpus (Zhu et al., 2015) is no longer freely distributed. We follow the suggestions from (Devlin et al., 2018) to crawl and collect Book-Corpus[4] on our own. The concatenation of two datasets contains roughly 3.4B words in total, which is comparable with the data corpus used in (Devlin et al., 2018). We randomly split documents into one training set and one validation set. The training-validation ratio for pre-training is 199:1.

We use `base` model configuration in our experiments. Similar to the translation task, we train the Pre-LN BERT without the warm-up stage and compare it with the Post-LN BERT. We follow the same hyper-parameter configuration in Devlin et al. (2018) to train the Post-LN BERT using 10k warm-up steps with $lr_{max} = 1e^{-4}$. For the Pre-LN BERT, we use linear learning rate decay starting from $3e^{-4}$ without the warm-up stage. We have tried to use a larger learning rate (such as $3e^{-4}$) for the Post-LN BERT but found the optimization diverged. All experiments are conducted on 32 NVIDIA Tesla P40 GPUs.

### 4.2 EXPERIMENT RESULTS

**Machine Translation**   We record the model checkpoints for every epoch during training and calculate the validation loss and BLEU score. The performance of the models at different checkpoints are plotted in Figure 4(a) - 4(d).

---

[4]https://www.smashwords.com

First, as we can see from the figure, the learning rate warm-up stage is not critical anymore for training the Pre-LN Transformer and the performance of the learned model is competitive. For example, on the IWSLT14 De-En task, the BLEU score and validation loss of the Pre-LN Transformer can achieve around 34 and 4, which are comparable with the performance of the Post-LN Transformer.

Second, the Pre-LN Transformer converges faster than the Post-LN Transformer using the same $\text{lr}_{max}$. On the IWSLT14 De-En task, the 9-th checkpoint of the Pre-LN Transformer achieves nearly the same performance (validation loss/BLEU score) as 15-th checkpoint of the Post-LN Transformer. Similar observations can be found in the WMT14 En-De task. The first model checkpoint of the Pre-LN Transformer can achieve a BLEU score near 20. As a comparison, the BLEU score of the first checkpoint of the Post-LN Transformer is less than 10.

Third, compared with RAdam, we find that the change of the position of layer normalization "dominates" the change of the optimizer. According to our experiments on the IWSLT14 De-En task, we can see that although RAdam trains the Post-LN Transformer well without the warm-up stage, it has little difference with Adam when training the Pre-LN Transformer.

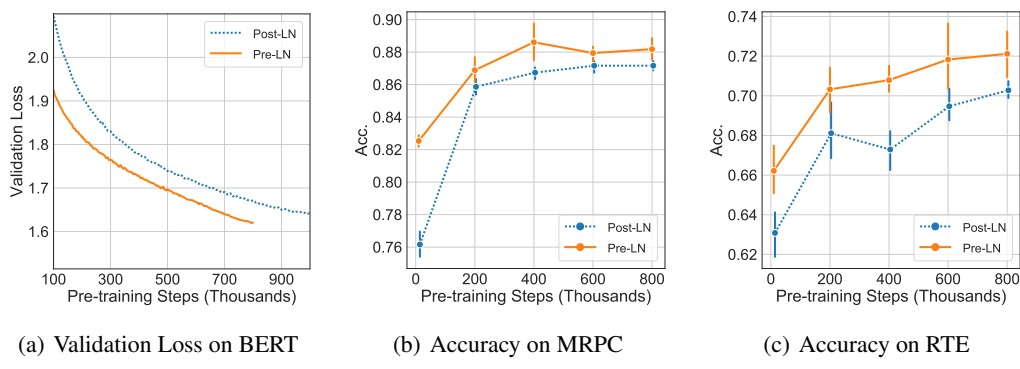

(a) Validation Loss on BERT       (b) Accuracy on MRPC       (c) Accuracy on RTE

Figure 5: Performances of the models on unsupervised pre-training (BERT) and downstream tasks

**Unsupervised Pre-training (BERT)** We record validation loss of the model checkpoints and plot them in Figure 5(a). Similar to the machine translation tasks, the learning rate warm-up stage can be removed for the Pre-LN model. The Pre-LN model can be trained faster. For example, the Post-LN model achieves 1.69 validation loss at 500k updates while the Pre-LN model achieves similar validation loss at 700k updates, which suggests there is a 40% speed-up rate. Note that $T_{warmup}$ (10k) is far less than the acceleration (200k) which suggests the Pre-LN Transformer is easier to optimize using larger learning rates. We also evaluate different model checkpoints on the downstream task MRPC and RTE (more details can be found in the appendix). The experiments results are plotted in Figure 5(b) and 5(c). We can see that the Pre-LN model also converges faster on the downstream tasks.

As a summary, the experiments on both machine translation and unsupervised pre-training tasks show that training the Pre-LN Transformer does not rely on the learning rate warm-up stage and can be trained much faster than the Post-LN Transformer.

## 5 CONCLUSION AND FUTURE WORK

In this paper, we study why the learning rate warm-up stage is important in training the Transformer and show that the location of layer normalization matters. We show that in the original Transformer, which locates the layer normalization outside the residual blocks, the expected gradients of the parameters near the output layer are large at the beginning of the optimization. This leads to an unstable training when using a large learning rate. We further show that the Transformer which locates the layer normalization inside the residual blocks, can be trained without the warm-up stage and converges much faster. In the future, we will investigate other strategies of positioning the layer normalization, as well as the advantage of layer normalization to the Transformer from a theoretical perspective.

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

# Appendix

## A    EXPERIMENTAL SETTINGS

### A.1    MACHINE TRANSLATION

**Experiment on Section 3**    The training/validation/test sets of the IWSLT14 German-to-English (De-En) task contain about 153K/7K/7K sentence pairs, respectively. We use a vocabulary of 10K tokens based on a joint source and target byte pair encoding (BPE) (Sennrich et al., 2015). All of our experiments use a Transformer architecture with a 6-layer encoder and 6-layer decoder. The size of embedding is set to 512, the size of hidden nodes in attention sub-layer and position-wise feed-forward network sub-layer are set to 512 and 1024, and the number of heads is set to 4. Label smoothed cross entropy is used as the objective function by setting $\epsilon = 0.1$ (Szegedy et al., 2016), and we apply dropout with a ratio 0.1. The batch size is set to be 4096 tokens. When we decode translation results from the model during inference, we set beam size as 5 and the length penalty as 1.2.

**Experiment on Section 4**    The configuration of IWLST14 De-En task is the same as in Section 3. For the WMT14 En-De task, we replicate the setup of (Vaswani et al., 2017), which consists of about 4.5M training parallel sentence pairs, and uses a 37K vocabulary based on a joint source and target BPE. Newstest2013 is used as the validation set, and Newstest2014 is used as the test set. One of the basic configurations of the Transformer architecture is the `base` setting, which consists of a 6-layer encoder and 6-layer decoder. The size of the hidden nodes and embeddings are set to 512. The number of heads is 8. Label smoothed cross entropy is used as the objective function by setting $\epsilon = 0.1$. The batch size is set to be 8192 tokens per GPU on 16 NVIDIA Tesla P40 GPUs.

### A.2    UNSUPERVISED PRETRAINING

We follow Devlin et al. (2018) to use English Wikipedia corpus and BookCorpus for the pre-training. As the dataset BookCorpus (Zhu et al., 2015) is no longer freely distributed. We follow the suggestions from Devlin et al. (2018) to crawl and collect BookCorpus[5] on our own. The concatenation of two datasets includes roughly 3.4B words in total, which is comparable with the data corpus used in Devlin et al. (2018). We first segment documents into sentences with Spacy[6]; Then, we normalize, lower-case, and tokenize texts using Moses (Koehn et al., 2007) and apply BPE(Sennrich et al., 2016). We randomly split documents into one training set and one validation set. The training-validation ratio for pre-training is 199:1.

The base model in Devlin et al. (2018) consists of 12 Transformer layers. The size of hidden nodes and embeddings are set to 768, and the number of heads is set to 12.

### A.3    GLUE DATASET

**MRPC**    The Microsoft Research Paraphrase Corpus (Dolan & Brockett, 2005) is a corpus of sentence pairs automatically extracted from online news sources, with human annotations for whether the sentences in the pair are semantically equivalent, and the task is to predict the equivalence. The performance is evaluated by the accuracy.

**RTE**    The Recognizing Textual Entailment (RTE) datasets come from a series of annual textual entailment challenges (Bentivogli et al., 2009). The task is to predict whether sentences in a sentence pair are entailment. The performance is evaluated by the accuracy.

**Fine-tuning on GLUE tasks**    We use the validation set for evaluation. To fine-tune the models, following Devlin et al. (2018); Liu et al. (2019b), we search the optimization hyper-parameters in a search space including different batch sizes (16/32), learning rates ($1e^{-5}$ - $1e^{-4}$) and number of epochs (3-8). We find that the validation accuracy are sensitive to random seeds, so we repeat

---

[5]https://www.smashwords.com
[6]https://spacy.io

fine-tuning on each task for 6 times using different random seeds and compute the 95% confidence interval of validation accuracy.

## B    PROOF OF LEMMA 1

*Proof.* Denote $X = (X_1, X_2, ..., X_d)$ in which $X_i$ are i.i.d. Gaussian random variables with distribution $N(0, \sigma^2)$. Denote $\rho_X(x)$ as the probability density function of $X_1$. Then $\mathbb{E}(\|\text{ReLU}(X)\|_2^2) = \sum_{i=1}^d \mathbb{E}[\text{ReLU}(X_i)^2] = \sum_{i=1}^d \mathbb{E}[\text{ReLU}(X_i)^2 | X_i \geq 0] \mathbb{P}(X_i \geq 0) = \frac{d}{2}\mathbb{E}[\text{ReLU}(X_1)^2 | X_1 \geq 0] = \frac{d}{2}\mathbb{E}[X_1^2 | X_1 \geq 0] = \frac{d}{2} \int_{-\infty}^{+\infty} x^2 \rho_{X|X>0}(x)dx = \frac{d}{2} \int_0^{+\infty} x^2 2\rho_X(x)dx = \frac{1}{2}\sigma^2 d.$ □

## C    PROOF OF LEMMA 2

*Proof.* At initialization, the layer normalization is computed as $\text{LN}(v) = \frac{v-\mu}{\sigma}$. It is easy to see that layer normalization at initialization projects any vector $v$ onto the $d-1$-sphere of radius $\sqrt{d}$ since $\|\text{LN}(v)\|_2^2 = \|\frac{v-\mu}{\sigma}\|_2^2 = \frac{\sum_{k=1}^d (v_k - \mu)^2}{\sigma^2} = d.$

We first estimate the expected $l_2$ norm of each intermediate output $x_{l,i}^{post,1}, \cdots, x_{l,i}^{post,5}$ for $l > 0$. Using Xavier initialization, the elements in $W^{V,l}$ are i.i.d. Gaussian random variables sampled from $N(0, 1/d)$. Since $\|x_{l,i}^{post}\|_2^2 = d$ by the definition of Layer Normalization when $l > 0$, we have

$$\mathbb{E}(\|x_{l,i}^{post,2}\|_2^2) = \mathbb{E}(\|x_{l,i}^{post}\|_2^2) + \mathbb{E}(\|x_{l,i}^{post,1}\|_2^2) + 2\mathbb{E}(x_{l,i}^{post,1} x_{l,i}^{post\top}) \tag{5}$$

$$= \mathbb{E}(\|x_{l,i}^{post}\|_2^2) + \mathbb{E}(\|x_{l,i}^{post,1}\|_2^2) + \frac{2}{n}\mathbb{E}(\sum_{j=1}^n x_{l,j}^{post} W^{V,l} x_{l,i}^{post\top}) \tag{6}$$

$$= \mathbb{E}(\|x_{l,i}^{post}\|_2^2) + \mathbb{E}(\|x_{l,i}^{post,1}\|_2^2) = \mathbb{E}(\|x_{l,i}^{post}\|_2^2) + \mathbb{E}(\|\frac{1}{n}\sum_{i=1}^n x_{l,i}^{post}\|_2^2) \leq 2d \tag{7}$$

and $\mathbb{E}(\|x_{l,i}^{post,2}\|_2^2) = \mathbb{E}(\|x_{l,i}^{post}\|_2^2) + \mathbb{E}(\|x_{l,i}^{post,1}\|_2^2) = \mathbb{E}(\|x_{l,i}^{post}\|_2^2) + \mathbb{E}(\|\frac{1}{n}\sum_{i=1}^n x_{l,i}^{post}\|_2^2) \geq \mathbb{E}(\|x_{l,i}^{post}\|_2^2) = d.$

Similarly, we have $\|x_{l,i}^{post,3}\|_2^2 = d$ by the definition of Layer Normalization. Again, for the ReLU activation function, the elements in $W^{1,l}$ and $W^{2,l}$ are i.i.d. Gaussian random variables sampled from $N(0, 1/d)$. According to Lemma 1, we have

$$\mathbb{E}(\|x_{l,i}^{post,4}\|_2^2) = \mathbb{E}(\|\text{ReLU}(x_{l,i}^{post,3} W^{1,l})W^{2,l}\|_2^2) \tag{8}$$

$$= \mathbb{E}(\mathbb{E}(\mathbb{E}(\|\text{ReLU}(x_{l,i}^{post,3} W^{1,l})W^{2,l}\|_2^2 | x_{l,i}^{post,3}, W^{1,l}) | x_{l,i}^{post,3})) \tag{9}$$

$$= \mathbb{E}(\mathbb{E}(\|\text{ReLU}(x_{l,i}^{post,3} W^{1,l})\|_2^2 | x_{l,i}^{post,3})) = \mathbb{E}(\frac{1}{2}\|x_{l,i}^{post,3}\|_2^2) = \frac{d}{2} \tag{10}$$

Based on this, we can estimate the scale of $\mathbb{E}(\|x_{l,i}^{post,5}\|_2^2)$ as follows.

$$\mathbb{E}(\|x_{l,i}^{post,5}\|_2^2) = \mathbb{E}(\|x_{l,i}^{post,3}\|_2^2) + \mathbb{E}(\|x_{l,i}^{post,4}\|_2^2) + 2\mathbb{E}(x_{l,i}^{post,3} x_{l,i}^{post,4\top}) \tag{11}$$

$$= \mathbb{E}(\|x_{l,i}^{post,3}\|_2^2) + \mathbb{E}(\|x_{l,i}^{post,4}\|_2^2) + \frac{2}{n}\mathbb{E}(\sum_{j=1}^n \text{ReLU}(x_{l,j}^{post,3} W^{1,l})W^{2,l} x_{l,i}^{post,3\top}) \tag{12}$$

$$= \mathbb{E}(\|x_{l,i}^{post,3}\|_2^2) + \mathbb{E}(\|x_{l,i}^{post,4}\|_2^2) = d + \frac{d}{2} = \frac{3}{2}d \tag{13}$$

Using similar technique we can bound $\mathbb{E}(\|x^{pre}_{l,i}\|^2_2)$ for the Pre-LN Transformer. Since

$$\mathbb{E}(\|x^{pre,3}_{l,i}\|^2_2) = \mathbb{E}(\|x^{pre}_{l,i}\|^2_2) + \mathbb{E}(\|x^{pre,2}_{l,i}\|^2_2) + 2\mathbb{E}(x^{pre,2}_{l,i}x^{pre\top}_{l,i}) \tag{14}$$

$$= \mathbb{E}(\|x^{pre}_{l,i}\|^2_2) + \mathbb{E}(\|x^{pre,2}_{l,i}\|^2_2) + \frac{2}{n}\mathbb{E}(\sum_{j=1}^{n} x^{pre,1}_{l,j}W^{V,l}x^{pre\top}_{l,i}) \tag{15}$$

$$= \mathbb{E}(\|x^{pre}_{l,i}\|^2_2) + \mathbb{E}(\|x^{pre,2}_{l,i}\|^2_2) = \mathbb{E}(\|x^{pre}_{l,i}\|^2_2) + \mathbb{E}(\|\frac{1}{n}\sum_{i=1}^{n} x^{pre,1}_{l,i}\|^2_2) \tag{16}$$

It is easy to see that we have $\mathbb{E}(\|x^{pre}_{l,i}\|^2_2) \leq \mathbb{E}(\|x^{pre,3}_{l,i}\|^2_2) \leq \mathbb{E}(\|x^{pre}_{l,i}\|^2_2) + d$. And similar to (11)-(13),

$$\mathbb{E}(\|x^{pre}_{l+1,i}\|^2_2) = \mathbb{E}(\|x^{pre,3}_{l,i}\|^2_2) + \mathbb{E}(\|x^{pre,5}_{l,i}\|^2_2) + 2\mathbb{E}(x^{pre,3}_{l,i}x^{pre,5\top}_{l,i}) \tag{17}$$

$$= \mathbb{E}(\|x^{pre,3}_{l,i}\|^2_2) + \mathbb{E}(\|x^{pre,5}_{l,i}\|^2_2) \tag{18}$$

$$= \mathbb{E}(\|x^{pre,3}_{l,i}\|^2_2) + \frac{1}{2}d \tag{19}$$

Combining both, we have $\mathbb{E}(\|x^{pre}_{l,i}\|^2_2) + \frac{1}{2}d \leq \mathbb{E}(\|x^{pre}_{l+1,i}\|^2_2) \leq \mathbb{E}(\|x^{pre}_{l,i}\|^2_2) + \frac{3}{2}d$. Then we have $(1 + \frac{l}{2})d \leq \mathbb{E}(\|x^{pre}_{l,i}\|^2_2) \leq (1 + \frac{3l}{2})d$ by induction.

$\square$

# D   PROOF OF LEMMA 3

The proof of Lemma 3 is based on Lemma 4:

**Lemma 4.** *Let $\alpha \in \mathbb{R}^d$ be a vector such that $\|\alpha\|_2 = 1$, then the eigenvalue of $I - \alpha^\top\alpha$ is either 1 or 0.*

*Proof.* Let $\{e_1, ..., e_d\}$ be unit vectors such that $e_1 = \alpha$ and $e_i \perp e_j$ for all $(i, j)$. Then we have $e_1(I - \alpha^\top\alpha) = e_1 - e_1\alpha^\top\alpha = e_1 - \alpha = 0$ and $e_i(I - \alpha^\top\alpha) = e_i - e_i\alpha^\top\alpha = e_i$ for $i \neq 1$. So $e_i$ are all the eigenvectors of $I - \alpha^\top\alpha$, and their corresponding eigenvalues are $(0, 1, 1, ..., 1)$. Hence we complete our proof. $\square$

*Proof of Lemma 3.* Denote $y = x(I - \frac{1}{d}\mathbf{1}^\top\mathbf{1})$, where $\mathbf{1} = (1, 1, ..., 1) \in \mathbb{R}^d$, then the layer normalization can be rewritten as

$$\text{LN}(x)_i = \frac{y_i}{\sqrt{\frac{1}{d}\sum_{j=1}^{d} y_j^2}} \tag{20}$$

We explicitly calculate the Jacobian of layer normalization as

$$\frac{\partial\text{LN}(x)_i}{\partial y_j} = \frac{\partial}{\partial y_j}(\frac{y_i}{\sqrt{\frac{1}{d}\sum_{k=1}^{n} y_k^2}}) = \frac{\delta_{ij}\sqrt{\frac{1}{d}\sum_{k=1}^{n} y_k^2} - y_i\frac{\frac{1}{d}y_j}{\sqrt{\frac{1}{d}\sum_{k=1}^{n} y_k^2}}}{\frac{1}{d}\sum_{k=1}^{n} y_k^2} \tag{21}$$

$$= \sqrt{d}\frac{\delta_{ij}\|y\|^2_2 - y_iy_j}{\|y\|^{\frac{3}{2}}_2} = \frac{\sqrt{d}}{\|y\|_2}(\delta_{ij} - \frac{y_iy_j}{\|y\|^2_2}) \tag{22}$$

where $\delta_{ij} = 1$ when $i = j$ and $\delta_{ij} = 0$ when $i \neq j$. In the matrix form,

$$\frac{\partial\text{LN}(x)}{\partial y} = \frac{\sqrt{d}}{\|y\|_2}(I - \frac{y^\top y}{\|y\|^2_2}) \tag{23}$$

and

$$\mathbf{J}_{LN}(x) = \frac{\partial \mathrm{LN}(x)}{\partial x} \tag{24}$$

$$= \frac{\partial \mathrm{LN}(x)}{\partial y} \frac{\partial y}{\partial x} \tag{25}$$

$$= \sqrt{d} \frac{1}{\|y\|_2} (I - \frac{y^\top y}{\|y\|_2^2})(I - \frac{1}{d}\mathbf{1}^\top \mathbf{1}). \tag{26}$$

Since the eigenvalue of the matrix $(I - \frac{y^\top y}{\|y\|_2^2})$ and $(I - \frac{1}{d}\mathbf{1}^\top \mathbf{1})$ are either 1 or 0 (by Lemma 4), we have $\|(I - \frac{y^\top y}{\|y\|_2^2})\|_2 = \mathcal{O}(1)$ and $\|(I - \frac{1}{d}\mathbf{1}^\top \mathbf{1})\|_2 = \mathcal{O}(1)$. So the spectral norm of $\mathbf{J}_{LN}(x)$ is

$$\|\mathbf{J}_{LN}(x)\|_2 = \mathcal{O}(\frac{\sqrt{d}}{\|y\|_2}) = \mathcal{O}(\frac{\sqrt{d}}{\|x\|_2}) \tag{27}$$

$\square$

## E    PROOF OF THEOREM 1

The proof of Theorem 1 is based on Lemma 5:

**Lemma 5.** *Let $Y$ be a random variable that is never larger than B. Then for all $a < B$,*

$$\Pr[Y \le a] \le \frac{\mathbb{E}[B - Y]}{B - a} \tag{28}$$

*Proof.* Let $X = B - Y$, then $X \ge 0$ and Markov's inequality tells us that

$$\Pr[X \ge B - a] \le \frac{\mathbb{E}[X]}{B - a} \tag{29}$$

Hence

$$\Pr[Y \le a] \le \frac{\mathbb{E}[B - Y]}{B - a} \tag{30}$$

$\square$

*Proof of Theorem 1.* We prove Theorem 1 by estimating each element of the gradient matrix. Namely, we will analyze $\frac{\partial \tilde{\mathcal{L}}}{\partial W_{pq}^{2,L}}$ for $p, q \in \{1, ..., d\}$. The loss of the post-LN Transformer can be written as

$$\tilde{\mathcal{L}}(x_{L+1,1}^{post}, ..., x_{L+1,n}^{post}) = \frac{1}{n} \sum_{i=1}^{n} \mathcal{L}(x_{L+1,i}^{post}) \tag{31}$$

Through back propagation, for each $i \in \{1, 2, ..., n\}$ the gradient of $\mathcal{L}(x_{L+1,i})$ with respect to the last layer's parameter $W^{2,L}$ in the post-LN setting can be written as:

$$\frac{\partial \mathcal{L}(x_{L+1,i}^{post})}{\partial W_{pq}^{2,L}} = \frac{\partial \mathcal{L}(x_{L+1,i}^{post})}{\partial x_{L+1,i}^{post}} \frac{\partial x_{L+1,i}^{post}}{\partial x_{L,i}^{post,5}} \frac{\partial x_{L,i}^{post,5}}{\partial x_{L,i}^{post,4}} \frac{\partial x_{L,i}^{post,4}}{\partial W_{pq}^{2,L}} \tag{32}$$

$$= \frac{\partial \mathcal{L}(x_{L+1,i}^{post})}{\partial x_{L+1,i}^{post}} \mathbf{J}_{LN}(x_{L,i}^{post,5}) \frac{\partial x_{L,i}^{post,4}}{\partial W_{pq}^{2,L}} \tag{33}$$

$$= \frac{\partial \mathcal{L}(x_{L+1,i}^{post})}{\partial x_{L+1,i}^{post}} \mathbf{J}_{LN}(x_{L,i}^{post,5})(0, 0, ..., [\mathrm{ReLU}(x_{L,i}^{post,3} W^{1,L})]_p, ..., 0)^\top \tag{34}$$

Here $[\text{ReLU}(x_{L,i}^{post,3}W^{1,L})]_p$ means the $p$-th element of $\text{ReLU}(x_{L,i}^{post,3}W^{1,L})$. So the absolute value of $\frac{\partial \mathcal{L}(x_{L+1,i}^{post})}{\partial W_{pq}^{2,L}}$ can be bounded by

$$|\frac{\partial \mathcal{L}(x_{L+1,i}^{post})}{\partial W_{pq}^{2,L}}| \leq \|\frac{\partial \mathcal{L}(x_{L+1,i}^{post})}{\partial x_{L+1,i}^{post}}\|_2 \|\mathbf{J}_{LN}(x_{L,i}^{post,5})\|_2 \|(0,0,...,[\text{ReLU}(x_{L,i}^{post,3}W^{1,L})]_p,...,0)^\top\|_2 \tag{35}$$

$$= \|\frac{\partial \mathcal{L}(x_{L+1,i}^{post})}{\partial x_{L+1,i}^{post}}\|_2 \|\mathbf{J}_{LN}(x_{L,i}^{post,5})\|_2 |[\text{ReLU}(x_{L,i}^{post,3}W^{1,L})]_p| \tag{36}$$

which implies

$$|\frac{\partial \mathcal{L}(x_{L+1,i}^{post})}{\partial W_{pq}^{2,L}}|^2 \leq \|\frac{\partial \mathcal{L}(x_{L+1,i}^{post})}{\partial x_{L+1,i}^{post}}\|_2^2 \|\mathbf{J}_{LN}(x_{L,i}^{post,5})\|_2^2 |[\text{ReLU}(x_{L,i}^{post,3}W^{1,L})]_p|^2 \tag{37}$$

Since we assume that all the derivatives are bounded, we have $\|\frac{\partial \mathcal{L}(x_{L+1,i}^{post})}{\partial x_{L+1,i}^{post}}\|_2^2 = \mathcal{O}(1)$. So

$$|\frac{\partial \mathcal{L}(x_{L+1,i}^{post})}{\partial W_{pq}^{2,L}}|^2 = \mathcal{O}\left([\|\mathbf{J}_{LN}(x_{L,i}^{post,5})\|_2^2 |[\text{ReLU}(x_{L,i}^{post,3}W^{1,L})]_p|^2\right]) \tag{38}$$

Since $\|x_{L,i}^{post,3}\|_2^2 = d$, $[x_{L,i}^{post,3}W^{1,L}]_p$ has distribution $N(0,1)$, using Chernoff bound we have

$$\Pr[|[x_{L,i}^{post,3}W^{1,L}]_p| \geq a_0] \leq \exp(-\frac{a_0^2}{2}).$$

So

$$\Pr[\text{ReLU}([x_{L,i}^{post,3}W^{1,L}]_p)^2 \geq 2\ln 100d] \leq \frac{0.01}{d}.$$

Thus with probability at least 0.99, for all $p = 1,2,...,d$ we have $\text{ReLU}([x_{L,i}^{post,3}W^{1,L}]_p)^2 \leq 2\ln 100d$.

Since with probability $1 - \delta(\epsilon)$, $\frac{|\|x_{L,i}^{post,5}\|_2^2 - \mathbb{E}\|x_{L,i}^{post,5}\|_2^2|}{\mathbb{E}\|x_{L,i}^{post,5}\|_2^2} \leq \epsilon$, we have $\|x_{L,i}^{post,5}\|_2^2 \leq (1+\epsilon)\mathbb{E}\|x_{L,i}^{post,5}\|_2^2$. Using Lemma 5, we have

$$\Pr[\|x_{L,i}^{post,5}\|_2^2 \leq \alpha_0\mathbb{E}\|x_{L,i}^{post,5}\|_2^2] \leq \frac{(1+\epsilon)\mathbb{E}\|x_{L,i}^{post,5}\|_2^2 - \mathbb{E}\|x_{L,i}^{post,5}\|_2^2}{(1+\epsilon-\alpha_0)\mathbb{E}\|x_{L,i}^{post,5}\|_2^2} = \frac{\epsilon}{1+\epsilon-\alpha_0} \tag{39}$$

for an arbitrary constant $\alpha_0 > 0$, which equals

$$\Pr[\|x_{L,i}^{post,5}\|_2^2 \geq \alpha_0\mathbb{E}\|x_{L,i}^{post,5}\|_2^2] \geq 1 - \frac{\epsilon}{1+\epsilon-\alpha_0} \tag{40}$$

So according to union bound, with probability at least $0.99 - \delta(\epsilon) - \frac{\epsilon}{1+\epsilon-\alpha_0}$ we have $|\frac{\partial \mathcal{L}(x_{L+1,i}^{post})}{\partial W_{pq}^{2,L}}|^2 = \mathcal{O}\left([\|\mathbf{J}_{LN}(x_{L,i}^{post,5})\|_2^2 |[\text{ReLU}(x_{L,i}^{post,3}W^{1,L})]_p|^2\right]) \leq \mathcal{O}(\frac{2d\ln 100d}{\|x_{L,i}^{post,5}\|_2^2}) \leq \mathcal{O}(\frac{d\ln d}{\alpha_0\mathbb{E}\|x_{L,i}^{post,5}\|_2^2}) = \mathcal{O}(\frac{\ln d}{\alpha_0})$. So we have

$$|\frac{\partial \tilde{\mathcal{L}}}{\partial W_{pq}^{2,L}}|^2 = |\frac{1}{n}\sum_{i=1}^{n} \frac{\partial \mathcal{L}(x_{L+1,i}^{post})}{\partial W_{pq}^{2,L}}|^2 \leq \frac{1}{n}\sum_{i=1}^{n}|\frac{\partial \mathcal{L}(x_{L+1,i}^{post})}{\partial W_{pq}^{2,L}}|^2 = \mathcal{O}(\frac{\ln d}{\alpha_0}) \tag{41}$$

and

$$\|\frac{\partial \tilde{\mathcal{L}}}{\partial W^{2,L}}\|_F = \sqrt{\sum_{p,q=1}^{d} |\frac{\partial \tilde{\mathcal{L}}}{\partial W_{pq}^{2,L}}|^2} = \mathcal{O}(\sqrt{\frac{d^2 \ln d}{\alpha_0}})$$

.

The loss of the pre-LN Transformer can be written as

$$\tilde{\mathcal{L}}(x_{Final,1}^{pre}, ..., x_{Final,n}^{pre}) = \frac{1}{n} \sum_{i=1}^{n} \mathcal{L}(x_{Final,i}^{pre}) \tag{42}$$

Using the same technique, in the pre-LN setting the gradient of $\mathcal{L}(x_{Final,i}^{pre})$ with respect to the last layer's parameter $W^{2,L}$ can be written as

$$\frac{\partial \mathcal{L}(x_{Final,i}^{pre})}{\partial W_{pq}^{2,L}} = \frac{\partial \mathcal{L}(x_{Final,i}^{pre})}{\partial x_{Final,i}^{pre}} \frac{\partial x_{Final,i}^{pre}}{\partial x_{L+1,i}^{pre}} \frac{\partial x_{L+1,i}^{pre}}{\partial x_{L,i}^{pre,5}} \frac{\partial x_{L,i}^{pre,5}}{\partial W_{pq}^{2,L}} \tag{43}$$

$$= \frac{\partial \mathcal{L}(x_{Final,i}^{pre})}{\partial x_{Final,i}^{pre}} \mathbf{J}_{LN}(x_{L+1,i}^{pre})(0, 0, ..., [\text{ReLU}(x_{L,i}^{pre,4} W^{1,L})]_p, ..., 0)^{\top} \tag{44}$$

So the absolute value of each component of the gradient is bounded by

$$|\frac{\partial \mathcal{L}(x_{Final,i}^{pre})}{\partial W_{pq}^{2,L}}| \leq \|\frac{\partial \mathcal{L}(x_{Final,i}^{pre})}{\partial x_{Final,i}^{pre}}\|_2 \|\mathbf{J}_{LN}(x_{L+1,i}^{pre})\|_2 \|(0, 0, ..., [\text{ReLU}(x_{L,i}^{pre,4} W^{1,L})]_p, ..., 0)\|_2 \tag{45}$$

$$= \|\frac{\partial \mathcal{L}(x_{Final,i}^{pre})}{\partial x_{Final,i}^{pre}}\|_2 \|\mathbf{J}_{LN}(x_{L+1,i}^{pre})\|_2 |[\text{ReLU}(x_{L,i}^{pre,4} W^{1,L})]_p| \tag{46}$$

Since $\|x_{L,i}^{pre,4}\|_2^2 = d$ and $[x_{L,i}^{pre,4} W^{1,L}]_p$ obeys distribution $N(0, 1)$, using Chernoff bound we have

$$\Pr[|[x_{L,i}^{pre,4} W^{1,L}]_p| \geq a_0] \leq \exp(-\frac{a_0^2}{2}).$$

So

$$\Pr[\text{ReLU}([x_{L,i}^{pre,4} W^{1,L}]_p)^2 \geq 2 \ln 100d] \leq \frac{0.01}{d}.$$

So with probability at least $0.99$, for all $p = 1, 2, ..., d$ we have $\text{ReLU}([x_{L,i}^{pre,4} W^{1,L}]_p)^2 \leq 2 \ln 100d$.

Since with probability $1 - \delta(\epsilon)$, $\frac{|\|x_{L+1,i}^{pre}\|_2^2 - \mathbb{E}\|x_{L+1,i}^{pre}\|_2^2|}{\mathbb{E}\|x_{L+1,i}^{pre}\|_2^2} \leq \epsilon$, we have $\|x_{L+1,i}^{pre}\|_2^2 \leq (1 + \epsilon)\mathbb{E}\|x_{L+1,i}^{pre}\|_2^2$. Using Lemma 5, we have

$$\Pr[\|x_{L+1,i}^{pre}\|_2^2 \leq \alpha_0 \mathbb{E}\|x_{L+1,i}^{pre}\|_2^2] \leq \frac{(1 + \epsilon)\mathbb{E}\|x_{L+1,i}^{pre}\|_2^2 - \mathbb{E}\|x_{L+1,i}^{pre}\|_2^2}{(1 + \epsilon - \alpha_0)\mathbb{E}\|x_{L+1,i}^{pre}\|_2^2} = \frac{\epsilon}{1 + \epsilon - \alpha_0} \tag{47}$$

which equals

$$\Pr[\|x_{L+1,i}^{pre}\|_2^2 \geq \alpha_0 \mathbb{E}\|x_{L+1,i}^{pre}\|_2^2] \geq 1 - \frac{\epsilon}{1 + \epsilon - \alpha_0} \tag{48}$$

According to union bound, with probability $0.99 - \delta(\epsilon) - \frac{\epsilon}{1+\epsilon-\alpha_0}$ we have $|\frac{\partial \mathcal{L}(x_{Final,i}^{pre})}{\partial W_{pq}^{2,L}}|^2 = \mathcal{O}\left(\|\mathbf{J}_{LN}(x_{L+1,i}^{pre})\|_2^2 |[\text{ReLU}(x_{L,i}^{pre,4} W^{1,L})]_p|^2\right) \leq \mathcal{O}(\frac{2d \ln 100d}{\|x_{L+1,i}^{pre}\|_2^2}) \leq \mathcal{O}(\frac{d \ln d}{\alpha_0 \mathbb{E}\|x_{L+1,i}^{pre}\|_2^2}) = \mathcal{O}(\frac{\ln d}{\alpha_0 L})$. So we have

$$|\frac{\partial \tilde{\mathcal{L}}}{\partial W_{pq}^{2,L}}|^2 = |\frac{1}{n} \sum_{i=1}^{n} \frac{\partial \mathcal{L}(x_{Final,i}^{pre})}{\partial W_{pq}^{2,L}}|^2 = \mathcal{O}(\frac{\ln d}{\alpha_0 L}) \tag{49}$$

Thus $\|\frac{\partial \tilde{\mathcal{L}}}{\partial W^{2,L}}\|_F = \sqrt{\sum_{p,q=1}^{d} |\frac{\partial \tilde{\mathcal{L}}}{\partial W_{pq}^{2,L}}|^2} \leq \mathcal{O}(\sqrt{\frac{d^2 \ln d}{\alpha_0 L}})$.

Take $\alpha_0 = \frac{1}{10}$, we have that with probability at least $0.99 - \delta(\epsilon) - \frac{\epsilon}{0.9+\epsilon}$, for the Post-LN Transformer we have $\|\frac{\partial \tilde{\mathcal{L}}}{\partial W^{2,L}}\|_F \leq \mathcal{O}(d\sqrt{\ln d})$ and for the Pre-LN Transformer we have $\|\frac{\partial \tilde{\mathcal{L}}}{\partial W^{2,L}}\|_F \leq \mathcal{O}(d\sqrt{\frac{\ln d}{L}})$ $\qquad \square$

## F    EXTENSION TO OTHER LAYERS

For simplicity, we denote $x_l = \text{Concat}(x_{l,1}, ..., x_{l,n}) \in \mathbb{R}^{nd}$ and $x_l^k = \text{Concat}(x_{l,1}^k, ..., x_{l,n}^k) \in \mathbb{R}^{nd}$ for $k = \{1, 2, 3, 4, 5\}$. Then in the Post-LN Transformer, the gradient of the parameters in the $l$-th layer (take $W^{2,l}$ as an example) can be written as

$$\frac{\partial \tilde{\mathcal{L}}}{\partial W^{2,l}} = \frac{\partial \tilde{\mathcal{L}}}{\partial x_{L+1}^{post}} \left( \prod_{j=l+1}^{L} \frac{\partial x_{j+1}^{post}}{\partial x_j^{post}} \right) \frac{\partial x_{l+1}^{post}}{\partial W^{2,l}}, \text{ where } \frac{\partial x_{j+1}^{post}}{\partial x_j^{post}} = \frac{\partial x_{j+1}^{post}}{\partial x_j^{post,5}} \frac{\partial x_j^{post,5}}{\partial x_j^{post,3}} \frac{\partial x_j^{post,3}}{\partial x_j^{post,2}} \frac{\partial x_j^{post,2}}{\partial x_j^{post}}.$$

The Jacobian matrices of the Post-LN Transformer layers are:

$$\frac{\partial x_{j+1}^{post}}{\partial x_j^{post,5}} = \begin{pmatrix} \mathbf{J}_{LN}(x_{j,1}^{post,5}) & & \\ & \ddots & \\ & & \mathbf{J}_{LN}(x_{j,n}^{post,5}) \end{pmatrix} \tag{50}$$

$$\frac{\partial x_j^{post,5}}{\partial x_j^{post,3}} = \begin{pmatrix} I & & \\ & \ddots & \\ & & I \end{pmatrix} + \begin{pmatrix} W^{2,j} & & \\ & \ddots & \\ & & W^{2,j} \end{pmatrix} \begin{pmatrix} \mathbf{J}_1^j & & \\ & \ddots & \\ & & \mathbf{J}_n^j \end{pmatrix} \begin{pmatrix} W^{1,l} & & \\ & \ddots & \\ & & W^{1,l} \end{pmatrix} \tag{51}$$

where $\mathbf{J}_i^j = \text{diag}\left( \sigma'\left( x_{j,i}^{post,3} \left( \mathbf{w}_1^{1,j} \right)^\top \right), ..., \sigma'\left( x_{j,i}^{post,3} \left( \mathbf{w}_d^{1,j} \right)^\top \right) \right) \in \mathbb{R}^{d \times d}$.

$$\frac{\partial x_j^{post,3}}{\partial x_j^{post,2}} = \begin{pmatrix} \mathbf{J}_{LN}(x_{j,1}^{post,2}) & & \\ & \ddots & \\ & & \mathbf{J}_{LN}(x_{j,n}^{post,2}) \end{pmatrix} \tag{52}$$

$$\frac{\partial x_j^{post,2}}{\partial x_j^{post}} = \begin{pmatrix} I & & \\ & \ddots & \\ & & I \end{pmatrix} + \begin{pmatrix} \frac{1}{n}W^{V,j} & \cdots & \frac{1}{n}W^{V,j} \\ \vdots & \ddots & \vdots \\ \frac{1}{n}W^{V,j} & \cdots & \frac{1}{n}W^{V,j} \end{pmatrix} \tag{53}$$

Using Hölder's inequality, we have

$$\mathbb{E}\|\frac{\partial x_{j+1}^{post}}{\partial x_j^{post}}\|_2 \leq \mathbb{E}\left[ \|\frac{\partial x_{j+1}^{post}}{\partial x_j^{post,5}}\|_2 \|\frac{\partial x_j^{post,5}}{\partial x_j^{post,3}}\|_2 \|\frac{\partial x_j^{post,3}}{\partial x_j^{post,2}}\|_2 \|\frac{\partial x_j^{post,2}}{\partial x_j^{post}}\|_2 \right] \tag{54}$$

$$\leq \sqrt{ \mathbb{E}\left[ \|\frac{\partial x_{j+1}}{\partial x_j^{post,5}}\|_2^2 \right] \mathbb{E}\left[ \|\frac{\partial x_j^{post,5}}{\partial x_j^{post,3}}\|_2^2 \|\frac{\partial x_j^{post,3}}{\partial x_j^{post,2}}\|_2^2 \|\frac{\partial x_j^{post,2}}{\partial x_j^{post}}\|_2^2 \right] } \tag{55}$$

Since $\frac{\partial x_{j+1}}{\partial x_j^{post,5}} = diag(\mathbf{J}_{LN}(x_{j,1}^{post,5}), ..., \mathbf{J}_{LN}(x_{j,n}^{post,5}))$, we have $\sqrt{ \mathbb{E}\left[ \|\frac{\partial x_{j+1}^{post}}{\partial x_j^{post,5}}\|_2^2 \right] } \approx \sqrt{ \mathbb{E}\frac{d}{\|x_{j,1}^{post,5}\|_2^2} } \approx \sqrt{\frac{2}{3}}$ when $\|x_{j,1}^{post,5}\|_2^2$ concentrates around its expectation $\mathbb{E}\|x_{j,1}^{post,5}\|_2^2$ which equals $\frac{3}{2}d$ according to Lemma 2. Therefore, when we estimate the norm of $\frac{\partial \tilde{\mathcal{L}}}{\partial W^{2,l}}$ for post-LN transformer, there exists a term $\mathcal{O}(\frac{2}{3}^{(L-l)/2})$, which exponentially decreases as $l$ goes smaller. Similarly, in the pre-LN Transformer, the gradient can be written as

$$\frac{\partial \tilde{\mathcal{L}}}{\partial W^{2,l}} = \frac{\partial \tilde{\mathcal{L}}}{\partial x_{Final}^{pre}} \frac{\partial x_{Final}^{pre}}{\partial x_{L+1}^{pre}} \left( \prod_{j=l+1}^{L} \frac{\partial x_{j+1}^{pre}}{\partial x_j^{pre}} \right) \frac{\partial x_{l+1}^{pre}}{\partial W^{V,l}}, \text{ where } \frac{\partial x_{j+1}^{pre}}{\partial x_j^{pre}} = \frac{\partial x_{j+1}^{pre}}{\partial x_j^{pre,3}} \frac{\partial x_j^{pre,3}}{\partial x_j^{pre}}.$$

The Jacobian matrices of the Pre-LN Transformer layers are:

$$\frac{\partial x_{j+1}^{pre}}{\partial x_j^{pre,3}} = \begin{pmatrix} I & & \\ & \ddots & \\ & & I \end{pmatrix} + \begin{pmatrix} W^{2,j} & & \\ & \ddots & \\ & & W^{2,j} \end{pmatrix} \begin{pmatrix} \mathbf{J}_1^{(h')} & & \\ & \ddots & \\ & & \mathbf{J}_n^{(h')} \end{pmatrix}$$

$$\begin{pmatrix} W^{1,j} & & \\ & \ddots & \\ & & W^{1,j} \end{pmatrix} \begin{pmatrix} \mathbf{J}_{LN}(x_{j,1}^{pre,3}) & & \\ & \ddots & \\ & & \mathbf{J}_{LN}(x_{j,n}^{pre,3}) \end{pmatrix}$$

$$\frac{\partial x_j^{pre,3}}{\partial x_j^{pre}} = \begin{pmatrix} I & & \\ & \ddots & \\ & & I \end{pmatrix} + \begin{pmatrix} \frac{1}{n}W^{V,j} & \cdots & \frac{1}{n}W^{V,j} \\ \vdots & \ddots & \vdots \\ \frac{1}{n}W^{V,j} & \cdots & \frac{1}{n}W^{V,j} \end{pmatrix} \begin{pmatrix} \mathbf{J}_{LN}(x_{j,1}^{pre}) & & \\ & \ddots & \\ & & \mathbf{J}_{LN}(x_{j,n}^{pre}) \end{pmatrix}$$

$$(56)$$

If $l$ is sufficiently large, the norm of $\mathbf{J}_{LN}(x_{j,i}^{pre})$ and $\mathbf{J}_{LN}(x_{j,i}^{pre,3})$ are very small (of order $\mathcal{O}(\frac{1}{\sqrt{j}})$) as $j$ is between $l+1$ and $L$, which means the eigenvalues of matrix $\frac{\partial x_{j+1}^{pre}}{\partial x_j^{pre,3}}$ and $\frac{\partial x_j^{pre,3}}{\partial x_j^{pre}}$ are close to 1. Then we can see that $\mathbb{E}\|\frac{\partial x_{j+1}^{pre}}{\partial x_j^{pre,3}}\|_2$ and $\mathbb{E}\|\frac{\partial x_j^{pre,3}}{\partial x_j^{pre}}\|_2$ are nearly 1, and the norm of $\frac{\partial \tilde{\mathcal{L}}}{\partial W^{2,l}}$ for pre-LN transformer is independent of $l$ when $l$ is large.

## G    EXAMPLES OF $(\epsilon, \delta)$-BOUNDED RANDOM VARIABLES

In this section we give an example of $(\epsilon, \delta)$-bounded random variable. This example comes from Example 2.5 in (Wainwright, 2019) and we give a short description below.

If $Z = (Z_1, ..., Z_n)$ is a Gaussian vector with distribution $N(0, I_n)$, then $Y = \|Z\|_2^2 = \sum_{k=1}^n Z_k^2$ has distribution $\chi_n^2$. And $\mathbb{E}Y = \sum_{k=1}^n \mathbb{E}Z_k^2 = n$

A random variable $X$ with mean $\mu = \mathbb{E}[X]$ is called *sub-exponential* if there are non-negative parameters $(\nu, \alpha)$ such that $\mathbb{E}[\exp(\lambda(X - \mu))] \leq \exp(\frac{\nu^2 \lambda^2}{2})$ for all $|\lambda| < \frac{1}{\alpha}$. The next proposition comes from Proposition 2.2 in (Wainwright, 2019).

**Proposition 1** (Sub-exponential tail bound). *Suppose that $X$ is sub-exponential with parameters $(\nu, \alpha)$. Then*

$$\mathbb{P}[X - \mu \geq t] \leq \begin{cases} \exp(-\frac{t^2}{2\nu^2}) & \text{if } 0 \leq t \leq \frac{\nu^2}{\alpha}, \text{ and} \\ \exp(-\frac{t}{2\alpha}) & \text{for } t > \frac{\nu^2}{\alpha} \end{cases} \tag{57}$$

*and from Example 2.5 in (Wainwright, 2019), the $\chi^2$ variable $Y$ is sub-exponential with parameters $(\nu, \alpha) = (2\sqrt{n}, 4)$. So we can derive the one-sided bound*

$$\mathbb{P}\left[Y - n \geq n\epsilon\right] \leq \exp(-n\epsilon^2/8), \quad \text{for all } \epsilon \in (0, 1) \tag{58}$$

*So $Y$ is $(\epsilon, \delta)$-bounded with $\epsilon \in (0, 1)$ and $\delta = \exp(-n\epsilon^2/8)$.*

## H    EMPIRICAL VERIFICATION OF THE THEORETICAL FINDINGS

As our theory is derived based on several simplifications of the problem, we conduct experiments to study whether our theoretical insights on the gradients are consistent with what we observe in real scenarios. We empirically study the gradients at initialization for both Post-LN/Pre-LN Transformer on the IWSLT14 De-En task. The general model and training configuration exactly follow Section 3.2. The experiments are repeated ten times using different random seeds.

**Empirical verification of concentration property**    Given an initialized model, we record the hidden states in the Post-LN/Pre-LN Transformer and find that the norm of the hidden states satisfies the concentration property ((0.1,0.125)-bounded).

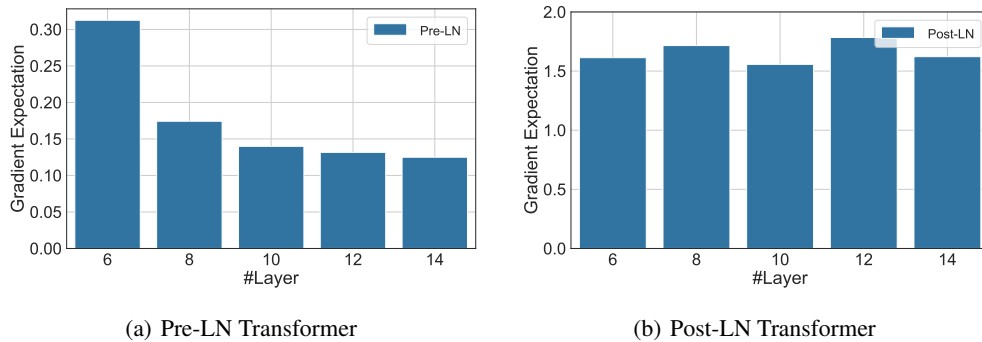

(a) Pre-LN Transformer        (b) Post-LN Transformer

Figure 6: Norm of expected gradients of $W^2$ in the last FFN sub-layer in different size of the Transformer architecture

**Empirical verification of Theorem 1**    In Theorem 1, the theory suggests that for any sizes of the Post-LN Transformer, the scale of the gradient norm in the last FFN sub-layer remains the same. On the contrary, the scale of the gradient norm in the last FFN sub-layer of the Pre-LN Transformer decreases as the size (total depth) of the model grows.

We conduct experiments to study the gradient norm in the last FFN sub-layer in different sizes of the Transformer architecture at initialization to verify Theorem 1. We train 6-6/8-8/10-10/12-12/14-14 Post-LN/Pre-LN Transformer models, and record the gradient norm of the final FFN layer in different Transformer models. The results are plotted in Figure 6. The x-axis is the size of the model, and the y-axis is the value of the gradient norm of $W^2$ in the final FFN sub-layer. It can be seen from the figure when the number of layers grows, the gradient norm remains in the Post-LN Transformer (around 1.6) and decreases in the Pre-LN Transformer. This observation is consistent well with our theory.

**Empirical verification of extended theory**    We record the gradient for each parameter for different mini-batches. For elements in a parameter matrix, we calculate their expected gradients and use the Frobenius norm of those values as the scale of the expected gradient of the matrix. Figure 3(a) and 3(b) shows those statistics for FFN sub-layers. The x-axis indexes different Transformer layers. It can be seen from the figure, the scale of the expected gradients grows along with the layer index for the Post-LN Transformer. On the contrary, the scale almost keeps the same for different layers in the Pre-LN Transformer. These observations are consistent with our theoretical findings.

Given the analysis above, we think our derived theory at the initialization stage is useful and consistent with the empirical studies above.

## I    LARGE GRADIENTS IN POST-LN TRANSFORMER HURTS THE OPTIMIZATION

Theoretically, we find that the gradients of the parameters near the output layers are very large for the Post-LN Transformer and suggest using large learning rates to those parameters makes the training unstable. To verify whether using small-step updates mitigates the issue, we conduct a set of experiments that follows the setting in Section 3.3. There are two simple ways of "small-step updates", gradient norm clipping, and using small learning rates.

**Experiments on using small learning rates**    We find using a very small but fixed learning rate can optimize the Post-LN Transformer (without the learning rate warm-up step) to a certain extent. We use a fixed learning rate of $1e^{-4}$ at the beginning of the optimization, which is much smaller than the $\mathrm{lr}_{max} = 1e^{-3}$ in the paper. Please note that as the learning rates during training are small, the training converges slowly, and this setting is not very practical in real large-scale tasks. We plot the validation curve together with other baseline approaches in Figure 7. We can see from the figure, the validation loss (pink curve) is around 4.3 in 27 epochs. This loss is much lower than that of the

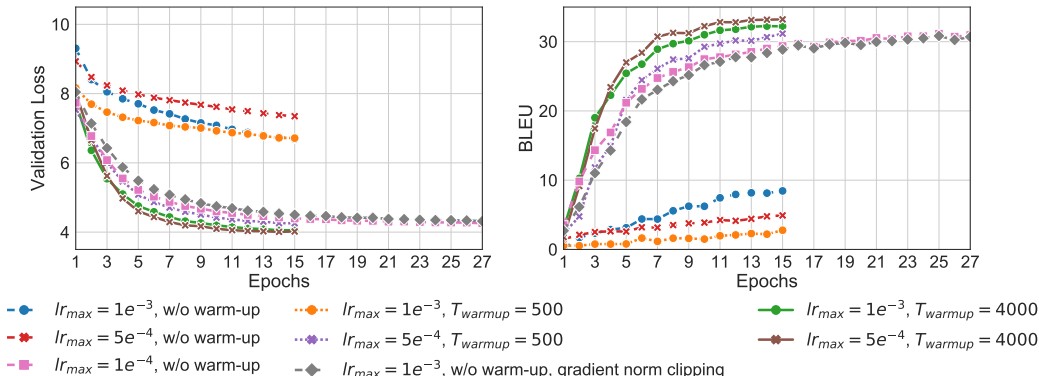

Figure 7: Performances of the models on the IWSLT14 De-En task.

Post-LN Transformer trained using a large learning rate (blue curve). But it is still worse than the SOTA performance (green curve).

**Experiments on using gradient norm clipping**  We find that using a small value of clip-norm can also optimize the Post-LN Transformer (without the learning rate warm-up stage). In the experiments, we find the gradient norm at initialization is about 5.00 and thus we clip the norm of the gradient update to 0.5. We plot the validation curve in Figure 7 (see gray curve). It can be seen from the figure that the performance is similar to the "small learning rate" experiment.

**Discussions**  We think the experiments above help to verify our conclusion. Using a small learning rate/clip norm mitigates the instability of updating models with large gradient values. Although the theory we prove is based on some simplifications of the problem, the theoretical insights help us understand the optimization of different networks in practice.

