# OpenReview forum: "On Layer Normalization in the Transformer Architecture"
_ICLR.cc/2020/Conference — Reject_

### Official Review · AnonReviewer2 · 2019-10-08
**Official Blind Review #2**

**Rating:** 6

**Review:**

Summary: The paper investigates the myth about layer normalization and learning rate warmup for the Transformer architecture. It shows, both theoretically and empirically, that putting the layer normalization in the residual blocks rather than between the residual blocks, could make a big difference to the scale of the gradient at the initialization stage.

Pros:
  + A well-written paper with a good organization; notations are clear.
  + The proof I checked seem correct (but I didn't check all of them).
  + Good experimental design that compares the pre-LN and post-LN Transformers in different settings/tasks.
  + Investigating the purpose and the theory behind using LN and learning rate warmup is a very interesting topic to me, as these modules are rarely used when training other kinds of deep nets (e.g., ConvNets, etc.).

I will detail the cons below, along with my other questions/concerns/issues.

----------------------------------------

Questions/concerns:

1. One of the two major concerns I have is the novelty of this paper in terms of its methodology and empirical value to the community. The Pre-LN setting of Transformers has already been widely used. For instance, Baevski et al. [1] and Child et al. [2] are both well-known works that have applied the pre-LN setting and achieved SOTA results on various very challenging benchmarks. These papers also reported that using layer normalization before self-attention brought "more effective training", which is one of the major empirical remarks that this paper made as well.

2. The second major concern I have is the connection the authors established between its theoretical findings and the empirical findings. While the post-LN Transformers may have larger gradient at higher-levels (as in, close-to-the-output levels), actually some (famous) prior works on Transformers have applied gradient clipping to their architecture, such as BERT [3] (https://github.com/google-research/bert/blob/master/optimization.py#L74), sparse transformer [2] and Transformer-XL [4]. But even when the gradient clipping is applied, learning rate warm-up still seems very helpful (and sometimes necessary), as was used in all of these works. Therefore, I think to further verify the theoretical hypotheses of the paper, the authors should at least also study whether (and to what degree) the very simple "gradient clipping" (or other gradient normalization techniques) solves the problem (which is a common solution to exploding gradients).

3. While Figure 3 is interesting to see, I don't think it verifies Theorem 1 exactly. What it verifies is the "extending to other layers/parameters" paragraph (i.e., the gradient scale decreases with layers). Did you try training post-LN Transformers and pre-LN Transformer with different # of layers from scratch (i.e., different L)? According to Thm. 1, I think we should expect to see a plot where post-LN gradient expectation remains at the same level for all L, and pre-LN gradually decreases with L.

4. The proofs are based on the core assumption that we are "at initialization" (e.g., you assumed W_V entries are sampled from N(0, 1/d), that W_Q=W_K=0, and that the input data are normally distributed x ~ N(0, \sigma^2 I_d)). How will the conclusion/derivation to change when these conditions are no longer met (e.g., after a few steps of warmup)? What do you expect to be the relationship between "the number of warmup steps" and solving the "gradient scale problem", which you proved on these assumptions?

======================================

Some minor issues that didn't impact the score:

1. You used \delta = e^{-d \epsilon^2 / 8} when discussing the tail bound, and later 3e^{-4}, 1e^{-4} for the learning rate. Just for notational consistency, maybe use 10^{-4} for the learning rate instead.

2. Referred to the wrong equation numbers (mentioned in another comment from me).

3. Appendix C: radius d -> radius \sqrt{d}

4. Appendix C: "Similarly, we have \| x_{l,i}^{post, 3} \|_2^2 = d" should have an expected value.

======================================

Despite the potential lack of novelty on method, I do think investigating these myths and instabilities of training Transformers is a very interesting direction to pursue. I think this paper can be improved with more experimental settings to verify the claim it proposes (i.e., on the gradient, which brings a lot of different things to analyze/study/fix here). I put the paper slightly above the acceptance borderline.



[1] https://arxiv.org/abs/1809.10853
[2] https://arxiv.org/abs/1904.10509
[3] https://arxiv.org/abs/1810.04805
[4] https://arxiv.org/abs/1901.02860

**Experience Assessment:**

I have published one or two papers in this area.

**Review Assessment: Checking Correctness Of Derivations And Theory:**

I assessed the sensibility of the derivations and theory.

**Review Assessment: Checking Correctness Of Experiments:**

I carefully checked the experiments.

**Review Assessment: Thoroughness In Paper Reading:**

I read the paper thoroughly.

---

> ### Author Response · Authors · 2019-11-10
> **Author Response**
>
> Dear AnonReviewer2,
>
> Thank you for the careful review and useful comments! Here are our responses to your questions and concerns.
>
> [Regarding Question 1]
>
> Although the Pre-LN Transformer has been used in previous works (we have already acknowledged this in Section 3.3), we still think our work is not incremental and has a novel understanding of the Transformer and its optimization.
>
> The main differences between our work and those previous works are in two aspects. First, as far as we know, in all previous works, Pre-LN Transformers are still trained with the **learning rate warm-up stage**, which is taken as granted from the Post-LN Transformer. Second, no previous works explain why and how the layer-normalization operation influences the optimization from a theoretical perspective, especially at initialization.
>
> Given such differences, our contribution is in two folds.
>
> First, different from all previous works, we are the first to formally study the gradients of the two Transformer at initialization from a theoretical perspective, and provide evidence to show why the learning rate warm-up stage is essential in training the Post-LN Transformer. The theory also suggests that the layer normalization matters, and the gradients in the Pre-LN Transformer are well-behaved.
>
> Second, we are the first to reveal that the **learning-rate warm-up stage can be removed** for training the Pre-LN Transformer, which simplifies the optimization process and improves the training efficiency. This finding is new to the community.
>
> Given the two points above, we think that our theoretical/empirical findings are novel. We hope the reviewer can re-evaluate our contribution.
>
> [Regarding Question 2]
>
> We thank R2 very much for pointing out the details in training BERT/Transformer-XL. We missed it as we just checked the papers, which didn't mention the gradient clipping. We further checked the codes of fairseq in PyTorch and found that the training of Roberta did not use the gradient clipping.
>
> Given the concerns about the gradient clipping, we conducted a new experiment to train a Post-LN Transformer translation model on the IWSLT14 De-En task. We trained the model without the learning rate warm-up stage and set the gradient clipping to be 1.0 (as in the BERT code). However, the experimental result showed that the model is poorly optimized, and the final BLEU score is less than 10, which suggests even with gradient clipping, the learning rate warm-up stage is still essential for training the Post-LN Transformer.
>
> We further study why this happens. We find that at initialization, the gradient norm (computed over all parameters) is about 5, which is not large. Therefore, the gradients of the parameters near the final layer after clipping can be still large. This may be one reason that the gradient clipping does not solve the problem.
>
> [Regarding Question 3]
>
> Thank you for pointing out this. This suggestion is quite useful to verify our theory. We conducted an experiment to train 6-6/8-8/10-10/12-12/14-14 Post-LN/Pre-LN Transformer translation models on the IWSLT14 De-En task, and record the gradient norm of the final FFN layer in different Transformer models at initialization.
>
> We plotted the results in Figure 6. The x-axis is the size of the model, and the y-axis is the value of the gradient norm of the final FFN layer. It can be seen from the figure when the number of layers grows, the gradient norm of the final FFN layer remains the same in the Post-LN model and decreases in the Pre-LN model. This observation is consistent well with our theory.
>
> [Regarding Question 4]
>
> We agree with the reviewer that understanding of the optimization process of the Transformer model is important and interesting. We are working on removing some assumptions, such as the zero initialization of W^Q and W^V. We hypothesize the theory will not change much. We are also super interested in what happens during the optimization of the Post-LN Transformer with the learning rate warm-up stage. However, giving an exact theoretical characterization of the optimization process for deep neural networks is very difficult, even for simple multi-layer perceptions. We will try our best.
>
>
>
> We thank the reviewer's carefully reading, and have corrected the typos and updated the version. We sincerely hope our explanations address the problem and concerns. We are also willing to answer other questions that you still have further.

---

> > ### Comment · AnonReviewer2 · 2019-11-10
> > **Further comments from Reviewer 2**
> >
> > Thank you for your response. I appreciate your effort in the additional studies and the detailed discussions. I've also read the review from & your rebuttals to R3 and R4.
> >
> > Some follow-up comments:
> >
> > 1) I'm still not entirely convinced of the novelty claim that you argued for. While it is indeed true that your work no longer take "learning rate warm-up" for granted, the actual methodology and the value of the it has been previously explored and widely used, as was also identified by R3 and R4 (and in practice, learning-rate warmup doesn't add much burden to implementation; but of course, I don't think it means exploring its cause is unimportant). Moreover, while you claimed that no previous works explain why and how LN influences the optimization from a theoretical perspective, your theory (strictly speaking) only applies **to the initialization stage**, with specifically $W_Q = W_K = 0$. This is still a pretty limited scope. I do acknowledge what you say, that giving exact theoretical formulations of general deep learning is very difficult; but meanwhile, I don't think it contradicts the issue of "limited scope" on a "previously explored & widely employed methodology".
> >
> > 2) Your observation on gradient clipping is interesting and I want to follow up further on that. I don't see why it's a "Therefore" between sentences "the gradient norm (...) is about 5, which is not large" and "the gradients (...) near the final layer after clipping can still be large." Doesn't GradNorm(last_layer) < GradNorm(all_params)? Moreover, doesn't your empirical observation that the gradient clipped model still needs learning-rate warm-up suggest that there are some problems deeper than just the gradient norm of the final layer?
> >
> > Plus, (though as I said above, I don't quite see the "therefore") if the final layer after clipping "can still be large", I think you can further try explicitly clipping the last (few) layers. This is possible in PyTorch, for instance, via "torch.nn.utils.clip_grad_norm_(model.last_layer.parameters(), ...)". I wonder if that fixes the problem you claimed in the paper. (Also, I think you can try even smaller clip value, like 0.25, which Transformer-XL used.)
> >
> > -----------------------------------------------
> >
> > I think an essential feeling I have is that the connection you claimed in the paper is vague. You showed that 1) "using Pre-LN" and "no need for learning-rate warm-up" coexisted; 2) "using Post-LN" and "definitely need learning-rate warm-up" coexisted; and 3) "Post-LN leads to large final-layer gradient norm while Pre-LN doesn't". But #1, #2, #3 together don't imply that "large final-layer gradient norm" is the cause of "definitely need learning-rate warm-up". This is why I think a detailed investigation over gradient clipping, which is a much more direct way of verifying this claim, is needed.

---

> > > ### Author Response · Authors · 2019-11-11
> > > **Further Author Response**
> > >
> > > Dear AnonReviewer2,
> > >
> > > Thank you for the quick feedback and suggestions! Your advice pushes us to take a deeper dive into the problem, which definitely makes our work more solid.
> > >
> > > In the paper, we find that the gradients near the output layer are large and suggest that ``using larger-step updates on such gradients leads to unstable optimization''. To verify whether using small-step updates mitigates the issue, we conduct a set of experiments with 6-6 Post-LN Transformer translation models on the IWSLT14 De-En task. There are two simple ways of ``small-step updates'', gradient norm clipping, and using small learning rates.
> > >
> > > [Experiments]
> > >
> > > 1. We find using a very small but fixed learning rate can optimize the Post-LN Transformer (without the learning rate warmup step) to a certain extent. We use a fixed learning rate of 1e-4 (corrected) during training, which is much smaller than the lr_max = 1e-3 in the paper. As can be expected, the training converges much slower. The validation loss is around 4.41 in 15 epochs. This loss is much lower than that of the Post-LN Transformer trained using a large learning rate (Figure 2.a). But it is still worse than the SOTA performance (e.g., 3.95). We also tried larger learning rates (1e-3,5e-4,2e-4), but the model is poorly optimized.
> > >
> > > 2. We find that the clip gradient norm 1.0 we used previously is too large. Using a much smaller value of clip_norm can optimize the Post-LN Transformer (without the learning rate warmup step). But the final performance is similar to the above experiment, and the convergence is very slow.
> > >
> > > We also tried to clip the gradient norm only for the last 1/2/3 layers as suggested and trained the model using fixed large learning rates. However, the results are not good. We hypothesize this is due to that gradient norm clipping provides small but biased gradients, it is hard to guarantee that the model is optimized towards a better loss when using mixed unbiased/biased gradients during training.
> > >
> > > [Discussions]
> > >
> > > We think the two sets of experiments above help to verify our conclusion. Using a small learning rate/clip_norm mitigates the instability of updating models with large gradient values. But both experiments show that the optimization converges slowly. We agree that the theory we prove is based on constraints and assumptions, but the theoretical insights help us explain optimization tricks and compare different networks in practice.
> > >
> > > We fully agree that in practice, learning-rate warmup doesn't add much burden to implementation. However, in the Post-LN Transformer, the final model performance is sensitive to the hyperparameters in the learning-rate warmup. It is time-consuming to tune these hyperparameters on different sizes of models using different sizes of datasets. We show that one doesn't need to take such heavy jobs as granted for Pre-LN Transformer: The training is easier and faster, with no lose (or even better) performance.
> > >
> > > We hope our extended experiments and analysis address your concerns, and we are willing to answer any questions.
> > >
> > > Thank you again for the suggestions.

---

> > > > ### Comment · AnonReviewer2 · 2019-11-11
> > > > **Add the extra information to revised version of the paper**
> > > >
> > > > Could you add these extra experiments to the paper (especially their convergence plots)? It's hard to tell what exactly is happening if you just say "the results are not good", or "the convergence is very slow". Comparison with Fig. 2(a) would be useful (but you can add these new empirical studies to the Appendix; your choice). I think these experiments are crucial anyway to make the paper more comprehensive (and will make it much stronger), should this paper be de-anonymized in the future.
> > > >
> > > > I also think that the claim of mixing unbiased/biased gradients is **too** hand-wavy. I would avoid marking that as the direct cause merely because the experiments didn't work out right for you (in fact, if you take a language model like Transformers and apply different clipping thresholds to the Transformer layers and the embedding layer--- which would be a "mix" of gradients with different levels of bias--- it would train just fine), without further study/investigation on the phenomenon. Instead, I think it exactly suggests the existence of a more profound problem than just the gradient norm of the last few layers.
> > > >
> > > > Also, if you are using smaller learning rates in certain scenarios, wouldn't it be fairer to train it for more epochs than when using larger learning rates? (E.g., imagine you are using lr=1e-7. I would bet it takes longer to train to reach a certain loss level than lr=1e-5...). But this is just a minor point :-)
> > > >
> > > > I look forward to re-reading the revision of the paper after the author rebuttal phase ends and before the reviewer/AC discussion. Thanks a lot to the authors for their efforts in running these experiments.

---

> > > > > ### Author Response · Authors · 2019-11-15
> > > > > **Paper revision**
> > > > >
> > > > > Dear AnonReviewer2,
> > > > >
> > > > > We have updated the paper with a new version. You can check Appendix H&I for the detailed experiments.
> > > > >
> > > > > Thanks for your suggestions.

---

### Official Review · AnonReviewer3 · 2019-10-23
**Official Blind Review #3**

**Rating:** 6

**Review:**


=========Update========
I appreciate the authors' response and additional appendix sections connecting theory and practice, verifying the claims about the final layer gradients scaling with L. I have also read the other reviews. I'm still not completely convinced that the multi-layer analysis well explains the reality, but because of the other contributions I am updating my score, which would be a 5/10 on the old ICLR scale.
=======================

This paper investigates the placement of Layer Normalization in the transformer architecture. The authors show that the Pre-LN placement leads to better behaved gradients as the network gets larger. This in turn allows them to remove the warmup stage of the learning rate schedule, leading to a faster and simpler training procedure. They examine a simplification of the attention layer analytically, and show a scaling with the number of layers that occurs with the Pre-LN placement but not the Post-LN placement. Finally, they demonstrate the effectiveness of the transformer changes both for machine translation and in BERT pretraining.

Even though the novelty here is limited, Pre-LN placement has been used in prior work, the potential for accelerating future research is large. In general I think the impact of this kind of research, exploring improvements to commonly used methods that add no additional complexity or even simplify, is underappreciated in the reviewing process. Still, I have some concerns about the relation between the analytic investigation of the gradient norms and the empirical results that are presented, and I am concerned that the analytical results are used to imply something stronger than they actually show.

At some level, it seems like the theoretical results have come along for the ride but do not clearly demonstrate that there is a problem with Post-LN and that this problem is fixed by switching to Pre-LN, or at least the relationship is not clearly explained. In the empirical study and central to the paper’s narrative is the fact that Post-LN normalization leads to a vanishing gradient problem where gradient in later layers is substantially larger in magnitude than in early layers at initialization, whereas the Pre-LN placement does not suffer from this problem. The point mentioned in the main text (theorem 1) is that the final FFN layer has gradient magnitude that scales at most like 1/sqrt(L) in the Pre-LN case and independent of L in the Post-LN case. This alone does not imply any of the observed empirical behavior because if all of the layers were scaled by this factor 1/sqrt(L) then the same problems would persist for the Pre-LN network. More relevant to the findings is how the scaling of the gradient changes throughout the layers which is examined in appendix section F, where the gradient norm of the Post-LN network can be upper bounded by an expression in which one of the terms scales approximately as (2/3)^(L-l)/2, whereas in the Pre-LN network the scaling is explained to be independent of layer index l. The connection between the expression that scales in the upper bound and the actual gradient norm is tenuous and there multiple places where the argument could break down (upper bound not being tight, scaling of other terms canceling out, validity of the simplifying assumptions being used). It would be useful to verify this sqrt(2/3) scaling on the data from the empirical study that is shown, is the decay shown in figure 3 geometric with this factor. Also the fact that the expectation near the bottom of page 20 is approximately 2/3 needs to be explained as it’s not obvious where that comes from.


**Experience Assessment:**

I have read many papers in this area.

**Review Assessment: Checking Correctness Of Derivations And Theory:**

I assessed the sensibility of the derivations and theory.

**Review Assessment: Checking Correctness Of Experiments:**

I carefully checked the experiments.

**Review Assessment: Thoroughness In Paper Reading:**

I read the paper at least twice and used my best judgement in assessing the paper.

---

> ### Author Response · Authors · 2019-11-10
> **Author Response**
>
> Dear AnonReviewer3,
>
> Thank you for the careful review and useful suggestions! According to the comments, we find you have general concerns about the relation between the theoretical findings and the empirical study. We address it in two aspects: Whether our theoretical findings can truly reflect the gradients at initialization in real practice, and whether our theoretical results can explain the optimization problems in training the Post-LN Transformer. Please kindly notify us if we misunderstand your concerns.
>
> 1. Whether the theory can truly reflect the gradients in real practice
>
> Given that our theoretical results are based on some assumptions (in Theorem 1) and relaxations or approximations (in Appendix F), we conduct more experiments to show that our theoretical findings are **consistent** with what we observe in real practice.
>
> First, Theorem 1 provides the relationship between the norm of the gradients in the last layer and the number of layers for both Pre/Post Transformer. We conduct an additional experiment in Appendix H to check whether it is correct in real tasks. In particular, we train 6-6/8-8/10-10/12-12/14-14 Post-LN/Pre-LN Transformer translation models on the IWSLT14 De-En task and record the gradient norm of the final FFN layer in different Transformer models. We plotted the results in Figure 6. The x-axis is the size of the model, and the y-axis is the value of the gradient norm of the final FFN layer. It can be seen from the figure when the number of layers grows, the gradient norm of the final FFN layer remains the same in the Post-LN model and decreases in the Pre-LN model. This observation is consistent well with Theorem 1.
>
> Second, our preliminary theoretical results in Appendix F are based on some approximated calculations which imply vanishing gradient in the Post-LN Transformer and stable gradients in the Pre-LN Transformer. We have conducted experiments to verify this as in Figure 3.
>
> Given the two experiments above, we think that our theoretical results can reflect what we observe at initialization in practice.
>
> 2. Whether the theory can explain the optimization problems in training the Post-LN Transformer
>
> We think our theoretical insights are useful to explain the optimization strategies. For the Post-LN Transformer, since the gradients are large in the upper layers and vanish along with the layers, using a large learning rate may cause the optimization unstable and lead to sub-optimal performance. This should be one reason that the learning rate warm-up stage is essential. Since the gradients are relatively stable and well-behaved in the Pre-LN Transformer, we empirically find that the learning rate warm-up stage can be removed, which reduces hyperparameter tuning and speed-up the training process.
>
> 3. On the term 2/3
>
> The term 2/3 comes from the fact that E[norm{x_{j,1}^{post, 5}}_2^2] = 3d/2 as proved in Lemma 2. We have stated it more clear in the latest version.
>
>
>
> As far as we know, this is the first work to touch the characterization of the gradients in the Transformer with layer normalization. We think our theoretical findings extend the knowledge of the Transformer models, and the removal of the learning rate warm-up stage has practical contributions.
>
> We sincerely hope that the reviewer can check our response and re-evaluate the contribution of the paper. It would be kind of you to upgrade your rating if you are satisfied with our response. Thank you for your re-consideration.

---

> > ### Author Response · Authors · 2019-11-15
> > **Thanks for your attention**
> >
> > Dear AnonReviewer3,
> >
> > We have uploaded a new version of the paper with more experiments. We believe we have addressed your concerns and clarified your points in the rebuttal/the updated paper. Do you have an updated assessment (or concerns) of our work? Thanks for your consideration.

---

### Official Review · AnonReviewer4 · 2019-10-29
**Official Blind Review #4**

**Rating:** 6

**Review:**

1. Specific problem tackled by the paper:

Moving the LayerNorm layer to be inside the residual connection in a stack of transformers can remove the need for learning rate warm up. This paper provides a theoretical motivation for doing this.

2. Motivation of the paper:

The authors motivate the problem clearly, performing experiments to demonstrate the problem. Their experiments provide good context for the problem they are solving and acts as a solid reference point for their (and other peoples' future) work. To be more convincing  the authors should have performed multiple runs and shown the standard deviations across runs.

3. Claims of the paper:

The authors claim that layer norm should be placed inside the residual connection (Pre-LN) rather than outside (Post-LN) it. The authors show, theoretically, that in a Post-LN transformer the magnitude of the gradients in a transformer scale with the number of layers and that the magnitude of the hidden states scale linearly with the layers that that they are output by. While in a Pre-LN transformer the magnitudes of the gradients and states are independent of the number of layers.

i.e. During training a Post-LN transformer is likely to have weaker gradients in the lower (closer to input layers) than at the output layers.

This theory is backed by an empirical study. It is good that these experiments are repeated ten times however the authors should show the standard deviations too.

The authors show machine translation results, demonstrating that using Pre-LN rather than Post-LN leads to faster convergence, however the models converge to the same result. However, there is benefit in not having to design a training schedule. Wang et al. have also shown the benefits of Pre-LN rather than Post-LN transformers for machine translation.

4. Decision (accept or reject) with one or two key reasons for this choice and reasons for the decision.

Weak accept.

Pros:
(1) The authors provide theory that supports the use of Pre-LN rather than Post-LN transformers. Using Pre-LN rather than Post-LN transformers may save a lot of time by avoiding hyper-parameter tuning, without loss in performance and this looks very easy to implement.
(2) The paper is well organised and the problem is very well motivated.
(3) The experiments are sufficient. They could be improved with repeats and by showing the standard deviations (as mentioned above).

Cons:
(1) This work is incremental. Wang et al. have already shown that Pre-LN transformers are better that Post-LN transformers when the network has many transformer layers, they explain theoretically why this is the case.
(2) While the organisation of the paper is good, the paper is not well written. The gramma is poor.
(3) The paper is very long for an incremental improvement.

5. Additional feedback with the aim to improve the paper.

[a] Ideally, it would be good to see repeats for Figure 2 and the standard deviations for Figure 3.
[b] Without reading the appendix it is not clear where the assumption that W^Q and W^K are zero is used. Making some connection with how this assumption relates to the lemmas would be useful. Additionally, you should explain what this means qualitatively, because this makes the assumption more acceptable. I am assuming that it means that the attention is uniform.
[c] In Lemma two you are comparing the magnitudes for the input in the Pre-LN and the output in the Post-LN transformer according to how x_{l,i}^post and x_{l, i}^pre are defined in Table 1.
[d] In Figure 3(b) the gradients are clearly decreasing with the number of layers, are there any comments on this? In the limit this could cause vanishing gradients?
[e] On the surface Figure 2 and 4 appear to contradict. Is the difference a result of using RAdam? If so, this should be made very clear. If not, why are the results contradictory?

More minor comments:
[1] Many gramma errors.
[2] Figure 1 is referenced before explaining what an FFN is. Figure 1 could also be enhanced by labelling Post-LN as previous work and Pre-LN as current work.
[3] MultiHeadAtt is not well defined. Multi-Head( ), Attention( ) and Head( ) each take three arguments, while MutliHeadAtt( ) takes two. It would be worth connecting these.
[4]"sub-layer --> ..." here sub-layer is not defined. Should this say self-attention sub-layer?
[5] Where does equation (4) come from? There should be a citation and/or explanation.
[6] The BLEU score is not defined (this could be with a footnote).
[7] Top of page 7: "As most of the parameters are initialized by Gaussian distributions" --> Using the word "most" is very vague. The authors should be specific about which parameters they are referring to.
[8] There is a good balance of equations in the main text with most of the proofs in the appendix.
[9] Inconsistent use of LN and LayerNorm, both are used.

**Experience Assessment:**

I have read many papers in this area.

**Review Assessment: Checking Correctness Of Derivations And Theory:**

I assessed the sensibility of the derivations and theory.

**Review Assessment: Checking Correctness Of Experiments:**

I assessed the sensibility of the experiments.

**Review Assessment: Thoroughness In Paper Reading:**

I read the paper at least twice and used my best judgement in assessing the paper.

---

> ### Author Response · Authors · 2019-11-10
> **Author Response**
>
> Dear AnonReviewer4,
>
> Thank you for the careful review and useful comments! Here are our responses to your questions and concerns.
>
> [Regarding Cons 1&3]
>
> Although the Pre-LN Transformer has been used in previous works (we have already acknowledged this in Section 3.3), we still think our work is not incremental and has a novel understanding of the Transformer and its optimization.
>
> The main differences between our work and those previous works are in two aspects. First, as far as we know, in all previous works, Pre-LN Transformers are still trained with the **learning rate warm-up stage**, which is taken as granted from the Post-LN Transformer. Second, no previous works explain why and how the layer-normalization operation influences the optimization from a theoretical perspective, especially at initialization.
>
> Given such differences, our contribution is in two folds.
>
> First, different from all previous works, we are the first to formally study the gradients of the two Transformer at initialization from a theoretical perspective, and provide evidence to show why the learning rate warm-up stage is essential in training the Post-LN Transformer. The theory also suggests that the layer normalization matters, and the gradients in the Pre-LN Transformer are well-behaved.
>
> Second, we are the first to reveal that the **learning-rate warm-up stage can be removed** for training the Pre-LN Transformer, which simplifies the optimization process and improves the training efficiency. This finding is new to the community.
>
> Given the two points above, we think that our theoretical/empirical findings are novel and worth a paper length of ten pages. We hope the reviewer can re-evaluate our contribution.
>
> [Regarding Cons 2]
>
> We thank the reviewer very much for pointing out the issues in our paper writing. We have revised the paper according to the suggestions in the **minor comments** and updated a new version.
>
> [Regarding the additional feedback [a]]
>
> Thank you for raising this question. We did repeat the experiment several times to confirm our empirical findings are correct (although the phenomena are concurrently observed by Liu. et al.). The observations are consistent using different random seeds, e.g., the difference between the max/min final BLEU  score in ten runs is less than 0.5.
>
> [Regarding the additional feedback [b]]
>
> Thanks for the suggestions. You are correct. The reason that we set W^Q and W^K to be zero matrices at initialization is that in this setting, the attention is uniform, which simplifies the calculation of the gradients. We have revised and updated the paper to give a sufficient explanation.
>
> [Regarding the additional feedback [c]]
>
> Thank you for raising this question. In the Pre-LN Transformer, the input in one layer is also the output of the previous layer. Therefore, we actually compared the outputs of the Pre-LN/Post-LN Transformer layers, more precisely, the outputs of the FFN sub-layer.
>
> [Regarding the additional feedback [d]]
>
> Yes. The gradient is likely to vanish in the Post-LN Transformer, which is also consistent with our theory (see the last paragraph on Page 20.)
>
> [Regarding the additional feedback [e]]
>
> In Figure 4, the Post-LN Transformer without the warm-up stage is trained with RAdam (see the legends). We have explicitly mentioned this in the ``machine translation'' paragraphs in Section 4.1&4.2.
>
>
>
> We sincerely hope our explanations address the problem and concerns. We are also willing to answer other questions that you still have further.

---

> > ### Comment · AnonReviewer4 · 2019-11-15
> > **[Regarding the additional feedback [d]]**
> >
> > Just to clarify I was saying that the gradients in Figure 3(b) are clearly decreasing with the number of layers for the Pre-LN Transformer as well. Do you have any explanation for this?
> >
> > Thank you.

---

> > > ### Author Response · Authors · 2019-11-15
> > > **Further Response**
> > >
> > > Dear AnonReviewer4,
> > >
> > > Thank you for the clarification. Fortunately, we did not miss the rebuttal deadline.
> > >
> > > Regarding your question, we quickly conduct an experiment for a 12-layer encoder/12-layer decoder Pre-LN Transformer model on the IWSLT De-En task. We calculate the norms of the expected gradient of W^1 and W^2 and record the values in the table below.
> > >
> > >  |Grad norm | L1  | L2  | L3   | L4  | L5  | L6   | L7  | L8  | L9  | L10| L11 | L12 |
> > > -------------------------------------------------------------------------------------------------------
> > >  |FFN W^2     |0.31|0.29|0.23|0.22|0.18|0.20|0.17|0.16|0.15|0.14|0.14|0.13|
> > > ----------------------------------------------------------------------------------------------------------
> > >  |FFN W^1     |0.16|0.14|0.12|0.11|0.11|0.10|0.09|0.09|0.09|0.07|0.07|0.07|
> > >
> > > First, we find you are correct. It can be seen from the table, the gradient norm decreases with the number of layers, which is similar to what we observed in the 6-6 Transformer in the paper. Second, the trend is not significant, e.g., compared to the gradient exponential decay in the Post-LN Transformer.
> > >
> > > The phenomenon is quite interesting, but currently (before the rebuttal deadline today), we cannot get a formal proof or a concrete explanation of what happens to these parameters as the Transformer is much more complicated than other networks. We would like to leave this, as well as a more in-depth understanding of the Transformer from an optimization perspective, as the future work.

---

### Comment · AnonReviewer2 · 2019-10-07
**Some questions**

Hi,

Some minor questions while reading the paper (I may be missing some assumptions, so please correct me if possible).

1) In your Eq. (6) -> (7), how did you eliminate the third term from the equation? If the covariance matrix of x_{l,i} and x_{l,j} is not diagonal, will the product still have an expected value of 0?

2) Right above Eq. (17), you said "similar to (16)-(21)". Do you mean (8)-(13) instead?

3) Did you clip your gradient in the experiments?

---

> ### Author Response · Authors · 2019-10-08
> **Author response**
>
> Dear Reviewer2:
>
> Thanks for your careful checking and quick feedback!
>
> 1.	We omit the subscript $l$ below for easy understanding. The expectation of the third term is 0 because
>
> a.	W is independent of x.
> b.	x_iWx_j^T is a scalar and is linear with respect to each element of W.
> c.	Each element of W is i.i.d sampled from a zero-centered distribution.
>
> Combining (a),(b),(c), we have  E[x_iWx_j^T]=\sum_{p,q} E[x_ip W_pq x_qj]= \sum_{p,q} E[W_pq]E[x_ip  x_qj]=0
>
> 2.	Sorry for the typos. You are correct. (16)-(21) should be (8)-(13). We will fix it.
>
> 3.	We followed (Vaswani 2017) and all other previous works and didn’t clip gradients in all experiments.
>
> Hope our clarifications can address your questions!

---

### Author Response · Authors · 2019-11-10
**General Author Response**


We thank all our reviewers for taking the time reading the paper and providing us with insightful comments and suggestions. We have responded to the questions, revised our paper accordingly, and updated a new version. We are willing to address any concerns from reviewers and meta reviewers further.

[Paper Updates]

- For notational consistency, use \exp instead of e^{} in the theorems and proofs.
- In Section 3.1, "sub-layer -> ..." is rephrased as "self-attention (FFN) sub-layer -> ...".
- In Section 3.2, we add references for Equation (4).
- In Section 3.3, we add more explanation on the assumptions.
- In Section 3.3, “most of the parameters are initialized by Gaussian distributions” is rephrased as "parameter matrices in self-attention sub-layers and FFN sub-layers are initialized by Gaussian distributions".
- In Appendix B, we correct the equation numbers.
- In Appendix C, radius d is corrected by radius \sqrt{d}.
- In Appendix F, we add discussions about where the coefficient 2/3 comes from.
- In Appendix H&I, we add experiments to verify the theory and support our explanations.

Thanks,
Paper227 Authors

---

### Decision · Program_Chairs · 2019-12-19

**Decision:**

Reject

**Comment:**

This paper investigates layer normalization and learning rate warmup in transformers, demonstrating that placing layer norm inside the residual connection (pre-LN) leads to better behaved gradients than post-LN placement. Doing so allows the learning rate warm-up stage to be removed, leading to faster training.

Reviewers were mildly positive about the submission, commenting on the interesting insight provided about transformers, as well as the clear, focused motivation and contribution.

However they also stated that it seem rather incremental of a contribution, as pre-LN placement has been introduced before, and found it confusingly written at times.

R2 clearly read it very closely, and had many detailed comments and discussions with authors and other reviewers. They had concerns about the relationship of this work with gradient clipping. The authors deserve credit for quickly investigating this in further experiments. Interestingly, the found that even with gradient clipping, post-LN models still needed the learning rate warm-up stage, although this issue went away with smaller clipped values or much lower learning rates. Overall, R2 appears to find the paper’s motivation very compelling, but the insights incomplete and not fully satisfactory, while all reviewers find the novelty rather limited.

I think a future submission that forges closer connections between the empirical findings and the theoretical interpretations would be of a great interest to the community, but in its current form is probably unsuitable for publication at ICLR 2020.